# OpenAUC: Towards AUC-Oriented Open-Set Recognition

**Zitai Wang**[1,2]    **Qianqian Xu**[3*]    **Zhiyong Yang**[4]
**Yuan He**[5]    **Xiaochun Cao**[6,1]    **Qingming Huang**[4,3,7,8*]

[1] SKLOIS, Institute of Information Engineering, CAS
[2] School of Cyber Security, University of Chinese Academy of Sciences
[3] Key Lab. of Intelligent Information Processing, Institute of Computing Tech., CAS
[4] School of Computer Science and Tech., University of Chinese Academy of Sciences
[5] Alibaba Group
[6] School of Cyber Science and Tech., Shenzhen Campus, Sun Yat-sen University
[7] BDKM, University of Chinese Academy of Sciences
[8] Peng Cheng Laboratory

`wangzitai@iie.ac.cn`    `xuqianqian@ict.ac.cn`
`yangzhiyong21@ucas.ac.cn`    `heyuan.hy@alibaba-inc.com`
`caoxiaochun@mail.sysu.edu.cn`    `qmhuang@ucas.ac.cn`

## Abstract

Traditional machine learning follows a close-set assumption that the training and test set share the same label space. While in many practical scenarios, it is inevitable that some test samples belong to unknown classes (open-set). To fix this issue, Open-Set Recognition (OSR), whose goal is to make correct predictions on both close-set samples and open-set samples, has attracted rising attention. In this direction, the vast majority of literature focuses on the pattern of open-set samples. However, how to evaluate model performance in this challenging task is still unsolved. In this paper, a systematic analysis reveals that most existing metrics are essentially inconsistent with the aforementioned goal of OSR: (1) For metrics extended from close-set classification, such as Open-set F-score, Youden's index, and Normalized Accuracy, a poor open-set prediction can escape from a low performance score with a superior close-set prediction. (2) Novelty detection AUC, which measures the ranking performance between close-set and open-set samples, ignores the close-set performance. To fix these issues, we propose a novel metric named OpenAUC. Compared with existing metrics, OpenAUC enjoys a concise pairwise formulation that evaluates open-set performance and close-set performance in a coupling manner. Further analysis shows that OpenAUC is free from the aforementioned inconsistency properties. Finally, an end-to-end learning method is proposed to minimize the OpenAUC risk, and the experimental results on popular benchmark datasets speak to its effectiveness.

## 1 Introduction

Traditional classification algorithms have achieved tremendous success under the close-set assumption that all the test classes are known during the training period. However, in many practical scenarios, it is inevitable that some test samples belong to none of the known classes. In this case, a close-set model will classify all the novel samples into known classes, inducing a significant performance degeneration. To fix this issue, Open-Set Recognition (OSR) has attracted rising attention in recent years [1, 2, 3, 4, 5, 6, 7, 8, 9, 10, 11, 12, 13, 14], where the model is required to not only (1) correctly classify the close-set samples but also (2) discriminate the open-set samples from the close-set ones.

---

*Corresponding authors.

36th Conference on Neural Information Processing Systems (NeurIPS 2022).

In this complicated setting, how to evaluate model performance becomes a challenging problem. Existing work has proposed several metrics, which fall into two categories:

The first direction extends traditional classification metrics to the open-set scenario. To this end, one should first extend the close-set confusion matrix with unknown classes, where a threshold decides whether the input sample belongs to the unknown classes. On top of this, **open-set F-score** [2, 9, 11, 12, 15, 16] summarizes the True Positive (TP), False Positive (FP), and False Negative (FN) performance of known classes. **Youden's index** [17] takes the sum of the True Positive Rate (TPR) and True Negative Rate (TNR) performance of known classes as the performance measure. Besides, **Normalized Accuracy** [15] summarizes the close-set accuracy and the open-set accuracy via a convex combination. Although it is intuitive to extend close-set metrics, we point out that these metrics are essentially inconsistent with the goal of OSR. Specifically, for open-set F-score and Youden's index, only the FP/FN performances of known classes evaluate the open-set performance implicitly. As a result, these metrics will encourage classifying close-set samples into the open-set to decrease the FN of known classes. Moreover, Normalized Accuracy encourages selecting the threshold classifying more open-set samples into known classes. In extreme cases, even a close-set model (*i.e.*, all the open-set samples are classified into known classes) can obtain a high performance on these metrics.

The second category regards OSR as a novelty detection problem [18, 19] with multiple known classes. Based on such observation, the Area Under ROC Curve (**AUC**) [20, 21], which measures the ranking performance between known classes and unknown classes, has become a popular metric [3, 4, 5, 6, 8, 10]. Compared with classification-based metrics, AUC is insensitive to the selection of threshold since it summarizes the True Positive Rate (TPR) performance for all possible thresholds. However, the limitation of AUC is also obvious: the close-set performance is ignored. A natural remedy is to adopt the close-set accuracy as a complementary metric [3]. However, what we expect is a model that can make correct predictions on close-set and open-set simultaneously. This decoupling strategy will induce a challenging multi-objective optimization problem and is also unfavorable to comparing the overall performances of different models. What's more, simply aggregating these two metrics will induce another inconsistency property.

In view of this, a natural question arises:

> *Whether there exists a numeric metric that is consistent with the goal of OSR?*

To answer this question, we propose a novel metric named **OpenAUC**. Specifically, the proposed metric enjoys a concise pairwise formulation, where each pair consists of a close-set sample and an open-set sample. For each pair, only if the close-set sample has been classified into the correct known class, OpenAUC will check whether the open-set sample is ranked higher than the close-set one. In this sense, OpenAUC evaluates the close-set performance and the open-set performance in a coupling manner, which is consistent with the goal of OSR. What's more, benefiting from the ranking operator, OpenAUC overcomes the sensitivity of the threshold, and further analysis shows that maximizing OpenAUC will guarantee a better open-set performance under a mild assumption on the threshold. Considering these advantages, we further establish an end-to-end learning method to maximize OpenAUC. Finally, extensive experiments conducted on multiple benchmark datasets validate the proposed metric and learning method. To sum up, the contribution of this paper is three-fold:

- We make a detailed analysis of existing metrics for OSR. The theoretical results show that existing metrics, including the classification-based ones and AUC, are essentially inconsistent with the goal of OSR due to their own limitations.
- A novel metric, named OpenAUC, is proposed. Benefiting from its concise formulation, further analysis shows that OpenAUC overcomes the limitations of existing metrics and thus is free from the inconsistency properties.
- An end-to-end learning method is proposed to optimize OpenAUC, and the empirical results on multiple benchmark datasets validate its effectiveness.

## 2  Preliminary

**Problem definition.** In open-set recognition, the training samples $\{z_i = (x_i, y_i)\}_{i=1}^n$ are drawn from a product space $\mathcal{Z}_k = \mathcal{X} \times \mathcal{Y}_k$, where $\mathcal{X}$ is the input space, and $\mathcal{Y}_k = \{1, \cdots, C\}$ is the

Table 1: The consistency analysis of existing metrics for OSR.

| Metric | P1 (close) | P2 (open) | P3 (threshold) | P4 (numeric) |
|---|---|---|---|---|
| Open-set F-score [15] | ✓ | ✗ | ✗ | ✓ |
| Youden's index [17] | ✓ | ✗ | ✗ | ✓ |
| Normalized Accuracy [15] | ✓ | ✓ | ✗ | ✓ |
| AUC [3] | ✗ | ✓ | ✓ | ✓ |
| The OSCR curve [4] | ✓ | ✓ | ✓ | ✗ |
| OpenAUC (Ours) | ✓ | ✓ | ✓ | ✓ |

label space of known classes. During the test period, some samples might belong to none of the known classes. For the sake of simplicity, all these samples can be allocated to one super unknown class. In other words, the open-set samples are drawn from a product space $\mathcal{Z}_u = \mathcal{X} \times \mathcal{Y}_u$, where $\mathcal{Y}_u = \{C + 1\}$ is the label space of unknown classes. To make predictions, OSR first requires a rejector $R = g_1 \circ r$ to judge whether an input sample comes from open-set, where $r : \mathcal{X} \to \mathbb{R}$ is the open-set score function, and $g_1 : \mathbb{R} \to \{0, 1\}$ is the open-set decision function. To be specific, an input $\boldsymbol{x}$ will be classified as an open-set sample if $r(\boldsymbol{x})$ is greater than a given threshold $t \in \mathbb{R}$. For the samples with $R(\boldsymbol{x}) = 0$, a classifier $h = g_2 \circ f$ is further required to make predictions on known classes, where $f : \mathcal{X} \to \mathbb{R}^C$ is the close-set score function and $g_2 : \mathbb{R}^C \to \mathcal{Y}_k$ is the close-set decision function. In view of this, a proper metric for OSR should enjoy the following properties:

- **(P1)** For close-set samples, the metric not only evaluates whether the open-set score function $r$ outputs low open-set scores but also requires that the classifier $h$ make correct predictions.

- **(P2)** For open-set samples, the metric should check whether the open-set score function $r$ outputs high open-set scores.

- **(P3)** The metric should be insensitive to the threshold $t$ because different ratios of open-set samples will induce different optimal thresholds, but such a ratio is unavailable during the training period.

- **(P4)** The metric should be a single numeric number to favor comparing the overall performances of different models.

**Roadmap.** Next, we first present a detailed analysis of existing metrics in Sec.3. The results show that these metrics are essentially inconsistent with the aforementioned properties, which are summarized in Tab.1. Furthermore, a novel metric named OpenAUC and its end-to-end learning method is proposed in Sec.4 to overcome the inconsistency of existing metrics.

## 3 Existing metrics for Open-set Recognition

Existing metrics for OSR fall into two categories: the classification-based ones and the novelty-detection ones. The first category extends existing classification metrics to the open-set scenario, while the second one regards OSR as a generalized novelty detection problem. We will present a detailed analysis of these metrics in the rest of this section.

### 3.1 Open-set F-score and Youden's Index

To extend classification metrics, one should first extend the confusion matrix with the unknown class. Let $\mathtt{TP}_i, \mathtt{TN}_i, \mathtt{FP}_i, \mathtt{FN}_i$ denote the True Positive (TP), True Negative (TN), False Positive (FP), False Negative (FN) of the class $i \in \mathcal{Y}_k \cup \mathcal{Y}_u$ under the given threshold $t$, respectively. Note that we omit the classifier $h$ and the rejector $R$ since there exists no ambiguity.

Open-set F-score [15] is a representative classification-based metric for OSR. Compared with its close-set counterpart, this metric evaluates the open-set performance via $\mathtt{FP}_i$ and $\mathtt{FN}_i$, where $i \in \mathcal{Y}_k$. To be specific, this metric summarizes $\mathtt{TP}_i, \mathtt{FP}_i, \mathtt{FN}_i$ of known classes by the harmonic mean of Precision and TPR (*i.e.*, Recall):

$$\mathtt{F\text{-}score} := 2 \times \frac{\mathtt{P}_k \times \mathtt{TPR}_k}{\mathtt{P}_k + \mathtt{TPR}_k}, \tag{1}$$

where

$$P_k := \frac{1}{C} \sum_{i=1}^{C} \frac{\text{TP}_i}{\text{TP}_i + \text{FP}_i}, \text{TPR}_k := \frac{1}{C} \sum_{i=1}^{C} \frac{\text{TP}_i}{\text{TP}_i + \text{FN}_i} \tag{2}$$

if one aggregates model performances in a macro manner, and

$$P_k := \frac{\sum_{i=1}^{C} \text{TP}_i}{\sum_{i=1}^{C} (\text{TP}_i + \text{FP}_i)}, \text{TPR}_k := \frac{\sum_{i=1}^{C} \text{TP}_i}{\sum_{i=1}^{C} (\text{TP}_i + \text{FN}_i)} \tag{3}$$

when model performances are summarized in a micro manner. Compared with open-set F-score, Youden's index additionally considers $\text{TN}_i$, where $i \in \mathcal{Y}_k$ [17, 22]:

$$J := \text{TPR}_k + \text{TNR}_k - 1, \tag{4}$$

where $\text{TNR}_k$ denotes the TNR of known classes. However, as illustrated in Prop.1, these two metrics suffer from an inconsistency property. Please refer to Appendix.B.1 for the proof.

**Proposition 1** (Inconsistency Property I). *Given a dataset $\mathcal{S}$ and a metric $M$ that is invariant to $\text{TP}_{C+1}$, $\text{FN}_{C+1}$ and $\text{FP}_{C+1}$, then for any $(h, R)$ such that $\sum_{i=1}^{C} \text{FP}_i(h, R) \geq \text{TP}_{C+1}(h, R)$, there exists $(\tilde{h}, \tilde{R})$ such that $M(\tilde{h}, \tilde{R}) = M(h, R)$ but $\text{TP}_{C+1}(\tilde{h}, \tilde{R}) = 0$.*

**Remark 1.** *If a metric $M$ suffers from the inconsistency property I, then for any $(h, R)$, we can construct $(\tilde{h}, \tilde{R})$ that performs as well as $(h, R)$ on $M$ but actually misclassifies all the open-set samples as known classes, which is inconsistent with (P2).*

**Remark 2.** $\sum_{i=1}^{C} \text{FP}_i(h, R) \geq \text{TP}_{C+1}(h, R)$ *is a mild condition. To be specific, when $\text{TP}_{C+1}$ is $\mathcal{O}(C)$, it only requires that $\text{FP}_i$ is $\mathcal{O}(1)$ for any $i \in \mathcal{Y}_k$. What's more, even if this condition does not hold, we still have $\text{TP}_{C+1}(\tilde{h}, \tilde{R}) < \text{TP}_{C+1}(h, R)$ as long as $\sum_{i=1}^{C} \text{FP}_i(h, R) \neq 0$.*

**Corollary 1.** *Open-set F-score and Youden's index both suffer from the inconsistency property I.*

## 3.2 Normalized Accuracy

Normalized Accuracy (NAcc) [15] summaries the accuracy performances on close-set and open-set:

$$\text{NAcc} := \lambda_{na}\text{AKS} + (1 - \lambda_{na})\text{AUS}, \tag{5}$$

where $\lambda_{na} \in (0, 1)$ is the balance constant, and

$$\text{AKS} := \frac{\sum_{i=1}^{C} [\text{TP}_i + \text{TN}_i]}{\sum_{i=1}^{C} [\text{TP}_i + \text{TN}_i + \text{FP}_i + \text{FN}_i]}, \text{AUS} := \frac{\text{TP}_{C+1}}{\text{TP}_{C+1} + \text{FP}_{C+1}} \tag{6}$$

are the Accuracy on Known and Unknown Samples (AKS, AUS), respectively. Since the close-set performance is explicitly involved, NAcc avoids the inconsistency property I. Ideally, if $\lambda_{na} = \mathbb{P}[y = C + 1]$, NAcc becomes exactly the close-set accuracy. However, it is generally hard to decide the balance constant $\lambda_{na}$ since we have no idea about the ratio of open-set samples in the test set. Besides, as shown in Prop.2, this metric suffers from another type of inconsistency property. Please refer to Appendix.B.2 for the proof.

**Proposition 2** (Inconsistency Property II). *Given a dataset $\mathcal{S}$, for any classifier-rejector pair $(h, R)$ such that $\sum_{i=1}^{C} \text{FN}_i(h, R) \geq \text{TP}_{C+1}(h, R)$ and $\text{TP}_{C+1}(h, R) > \text{FP}_{C+1}(h, R)$, there exists $(\tilde{h}, \tilde{R})$ such that $\text{NAcc}(\tilde{h}, \tilde{R}) > \text{NAcc}(h, R)$ but $\text{TP}_{C+1}(\tilde{h}, \tilde{R}) = 0$.*

**Remark 3.** *For any $(h, R)$, we can construct $(\tilde{h}, \tilde{R})$ such that $\text{NAcc}(\tilde{h}, \tilde{R}) > \text{NAcc}(h, R)$ but actually misclassifies all the open-set samples to known classes. In other words, NAcc encourages selecting a threshold that classifies more open-set samples to known classes, which is inconsistent with (P3).*

**Remark 4.** *Similar to the condition in Prop.1, $\sum_{i=1}^{C} \text{FN}_i(h, R) \geq \text{TP}_{C+1}(h, R)$ is a mild condition. And $\text{TP}_{C+1}(h, R) > \text{FP}_{C+1}(h, R)$ is also mild since it is a basic requirement for open-set models.*

## 3.3 The Area Under the ROC Curve (AUC)

When regarding OSR as a novelty detection problem, we no longer need to extend the confusion matrix, and AUC [3] comes into play. Traditional AUC is a popular metric for the imbalanced binary

classification problem since it is insensitive to the label distribution [23, 24]. To extend this metric to OSR, one has to allocate all the known classes to one super known class, which is the same as the label space of unknown classes. Let $\text{TPR}_s, \text{FPR}_s$ denote the TPR and FPR of the super known class, respectively. Then, AUC is defined as the area under the $\text{TPR}_s$-$\text{FPR}_s$ curve:

$$\text{AUC} := \int_{-\infty}^{+\infty} \text{TPR}_s(\text{FPR}_s^{-1}(t)) \, dt. \tag{7}$$

If we assume that there exist no ties in $r$, AUC will enjoy a much simpler formulation [23]:

$$\text{AUC} = \mathop{\mathbb{E}}_{\substack{z_k \sim D_k \\ z_u \sim D_u}} \left[ \mathbf{1} \left[ r(\boldsymbol{x}_u) > r(\boldsymbol{x}_k) \right] \right], \tag{8}$$

where $D_k, D_u$ are the distribution of known and unknown classes, respectively. In other words, AUC equals the probability that the close-set score function $r$ ranks an open-set sample higher than a close-set one. Compared with classification-based metrics, AUC summarizes the model performance under different thresholds, and the pairwise formulation (8) requires no thresholds. However, its limitation is also obvious: **the close-set performance is ignored**, which is inconsistent with **(P1)**. To address this issue, one can adopt the close-set accuracy, denoted by $\text{Acc}_k$, as a complementary metric. In other words, this strategy divides OSR into two traditional tasks: multiclass classification and novelty detection. Although intuitive, it is also inconsistent with the goal of OSR since the open-set performance and the close-set performance are evaluated in a decoupling manner. To be specific, when $R(\boldsymbol{x}) = 1$ for a close-set sample $\boldsymbol{x}$, even if $h$ makes a correct prediction on $\boldsymbol{x}$, which will improve the $\text{Acc}_k$ performance, $(R, h)$ will misclassify $\boldsymbol{x}$ to the unknown class. In this case, $\text{Acc}_k$ is inconsistent with the actual model performance. What's more, simply aggregating the two metrics will induce another inconsistency property, whose proof is presented in Appendix.B.3:

**Proposition 3** (Inconsistency Property III). *Given a dataset $\mathcal{S}$, for any $(h, r)$ satisfying $\text{Acc}_k, \text{AUC} \neq 1$, there exists $(\tilde{h}, \tilde{r})$ that performs worse on the OSR task but satisfies:*

$$agg(\text{Acc}_k(\tilde{h}), \text{AUC}(\tilde{r})) = agg(\text{Acc}_k(h), \text{AUC}(r)),$$

*where $agg : \mathbb{R} \times \mathbb{R} \to \mathbb{R}$ is the aggregation function.*

**Corollary 2.** *The following metrics suffer from the inconsistency property III: (1) The product of the close-set accuracy and AUC, i.e., $\text{Acc}_k \cdot \text{AUC}$, (2) the summation of the close-set accuracy and AUC, i.e., $\text{Acc}_k + \text{AUC}$, and (3) the pointwise summation of the close-set accuracy and AUC:*

$$\text{Acc}_k \oplus \text{AUC} := \mathop{\mathbb{E}}_{\substack{z_k \sim D_k \\ z_u \sim D_u}} \left[ \mathbf{1} \left[ y_k = h(\boldsymbol{x}_k) \right] + \mathbf{1} \left[ r(\boldsymbol{x}_u) > r(\boldsymbol{x}_k) \right] \right].$$

## 4 OpenAUC: a novel metric for Open-set Recognition

Considering the limitations of existing metrics, we present a novel metric in this section. To this end, we first review a non-numeric metric named the OSCR curve.

### 4.1 Open Set Classification Rate (OSCR)

The Open Set Classification Rate (OSCR) curve [4] is an adaptation of the Detection and Identification Rate (DIR) curve that is commonly used in open-set face recognition [25]. In this curve, $f(\boldsymbol{x})_c$ and $r(\boldsymbol{x})$ are assumed to be proportional to $\mathbb{P}[y = c|\boldsymbol{x}]$ and $1/\max_{c \in \mathcal{Y}_k} f(\boldsymbol{x})_c$, respectively. Then, x-axis is defined as the FPR performance on open-set samples, while y-axis is the Correct Classification Rate (CCR) defined on the close-set:

$$\text{CCR}(t) := \frac{1}{N_k} \sum_{i=1}^{N_k} \mathbf{1} \left[ y_i = \arg\max_{c \in \mathcal{Y}_k} f(\boldsymbol{x}_i)_c \right] \cdot \mathbf{1} \left[ f(\boldsymbol{x}_i)_{y_i} > t \right], \tag{9}$$

where $N_k$ is the number of close-set samples, and $\mathbf{1}[\mathcal{A}]$ is the indicator function of an event $\mathcal{A}$. Compared with AUC, the OSCR curve considers the close-set performance via CCR. Meanwhile, as FPR varies from 0 to 1, this curve will present the CCR performance under different thresholds.

Compared with the aforementioned metrics, the OSCR curve contains richer information and allows comparing model performances at different operating points. Even though, our goal is to optimize

the overall performance of the curve. Hence, it is necessary to find a numeric metric that aggregates the information of the entire curve, such that the models can be trained by optimizing the loss of the metric. To this end, [13] and [14] estimate the area under the OSCR curve by directly calculating the numeric integral with histograms. However, this number is hard to optimize due to multiple non-differential operators such as ranking and counting. Moreover, the assumptions on $(f, r)$ also limit the application of this metric. For example, considering that $\mathcal{Z}_u = \neg\mathcal{Z}_1 \cap \neg\mathcal{Z}_2 \cap \cdots \cap \neg\mathcal{Z}_C$, [13, 26] propose to model potential unknown space via the prototypes of $\{\neg\mathcal{Z}_c\}_{c=1}^C$. Inspired by this, we adopt the following definition, where the aforementioned assumptions on $(f, r)$ do not hold:

**Definition 1.** *Let $f(\boldsymbol{x})_c$ denote the probability that the input $\boldsymbol{x}$ does not belong to the class $c$, that is, $\forall c \in \mathcal{Y}_k, f(\boldsymbol{x})_c \propto \mathbb{P}[y \neq c|\boldsymbol{x}]$. In this case, $r(\boldsymbol{x})$ is defined as $\min_{k \in \mathcal{Y}_k} f(\boldsymbol{x})_k$ since given an open-set sample $\boldsymbol{x}$, a well-trained model tends to output a large $f(\boldsymbol{x})_k$ for any $k \in \mathcal{Y}_k$.*

### 4.2 The Definition of OpenAUC

To overcome the limitations of the OSCR curve, we first remove the assumptions on $(h, r)$ and present a generalized formulation of open-set FPR and CCR, denoted by OFPR and Conditional OTPR (COTPR), respectively:

$$\texttt{OFPR}(t) := \mathop{\mathbb{E}}_{\boldsymbol{z} \sim D_u} \left[\mathbf{1}\left[r(\boldsymbol{x}) \leq t\right]\right], \texttt{COTPR}(t) := \mathop{\mathbb{E}}_{\boldsymbol{z} \sim D_k} \left[\mathbf{1}\left[r(\boldsymbol{x}) \leq t, y = h(\boldsymbol{x})\right]\right], \quad (10)$$

In other words, OFPR represents the probability that $r$ misclassifies an open-set sample to known classes, and COTPR equals the probability that $(h, r)$ makes a correct prediction on a close-set sample. Then, OpenAUC is defined as the area under the OFPR–COTPR curve:

$$\texttt{OpenAUC} := \int_{-\infty}^{+\infty} \texttt{COTPR}(\texttt{OFPR}^{-1}(t)) \, dt. \quad (11)$$

However, this integral formulation is still hard to calculate. To fix this issue, we present the following reformulation, whose proof is presented in Appendix.C.1.

**Proposition 4.** *Given $(h, r)$ and a sample pair $(\boldsymbol{z}_k, \boldsymbol{z}_u), \boldsymbol{z}_k \in \mathcal{Z}_k, \boldsymbol{z}_u \in \mathcal{Z}_u$, OpenAUC equals the probability that $h$ makes correct prediction on $\boldsymbol{z}_k$ and $r$ ranks $\boldsymbol{z}_u$ higher than $\boldsymbol{z}_k$:*

$$\texttt{OpenAUC} = \mathop{\mathbb{E}}_{\substack{\boldsymbol{z}_k \sim D_k \\ \boldsymbol{z}_u \sim D_u}} \Big[ \underbrace{\mathbf{1}\left[y_k = h(\boldsymbol{x}_k)\right]}_{(a)} \cdot \underbrace{\mathbf{1}\left[r(\boldsymbol{x}_u) > r(\boldsymbol{x}_k)\right]}_{(b)} \Big]. \quad (12)$$

**Remark 5.** *First, for close-set samples, term (a) and term (b) require that $h$ makes correct predictions and $r$ outputs low open-set scores, which is consistent with **(P1)**. Second, term (b) requires $r$ to rank open-set samples higher than the close ones, which is consistent with **(P2)** and **(P3)**. Last but not least, the product operator guarantees that OpenAUC evaluates the close-set performance and the open-set performance in a coupling manner, which is essentially different from the metrics mentioned in Prop.3.*

Compared with existing numeric metrics, OpenAUC is free from the aforementioned inconsistency properties, and we will make a detailed discussion in Sec.4.3. Moreover, Compared with the OSCR curve, OpenAUC enjoys a concise formulation, based on which we can design a differentiable objective function for Empirical Risk Minimization (ERM) in Sec.4.4.

### 4.3 OpenAUC vs. Existing Metrics for OSR

Essentially, inconsistency property I is induced by the fact that exchanging the predictions of open-set samples and close-set samples will not lead to a larger error on these metrics. However, such an exchange at least breaks a sample pair that has been correctly ranked by $r$, while the close-set accuracy will not be improved. Thus, we have the following proposition, whose details are presented in Appendix.C.2.

**Proposition 5.** *Given a sample pair $((\boldsymbol{x}_1, C + 1), (\boldsymbol{x}_2, y_2))$, where $y_2 \neq C + 1$, for any $(h, r)$ such that $R(\boldsymbol{x}_1) = 1, R(\boldsymbol{x}_2) = 0, h(\boldsymbol{x}_2) \neq y_2$, if $(\tilde{h}, \tilde{r})$ makes the same predictions as $(h, r)$ expect that $\tilde{R}(\boldsymbol{x}_1) = 0, \tilde{h}(\boldsymbol{x}_1) = h(\boldsymbol{x}_2)$ and $\tilde{R}(\boldsymbol{x}_2) = 1$, we have $\texttt{OpenAUC}(\tilde{h}, \tilde{r}) < \texttt{OpenAUC}(h, r)$.*

**Remark 6.** *If we construct $(\tilde{h}, \tilde{r})$ in the way that leads to the inconsistency property I, $(\tilde{h}, \tilde{r})$ will perform inferior to $(h, r)$ on OpenAUC, that is, OpenAUC is free from the inconsistency property I.*

Besides, the following proposition reveals that optimizing OpenAUC will guarantee a lower bound of $\text{TPR}_{C+1}$, which helps avoid the inconsistency property II. Please refer to Appendix.C.3 for the proof.

**Proposition 6.** *Given a dataset $\mathcal{S}$, for any $(f, r)$ such that $\texttt{OpenAUC} = k$ and any threshold $t_{C+1}$ such that $\texttt{FPR}_{C+1} = a \neq 0$, we have $\texttt{TPR}_{C+1} \geq 1 - (1-k)/a$.*

**Remark 7.** *It is clear that $\texttt{TPR}_{C+1} > 0$ when $a > 1 - k$. As we optimize OpenAUC, that is, $k \to 1$, it will become easier to select a threshold $t_{C+1}$ such that $a > 1 - k$. In other words, OpenAUC is free from the inconsistency property II under a mild condition.*

Moreover, inconsistency property III is induced by the fact that $agg(\texttt{Acc}_k, \texttt{AUC})$ evaluates the close-set performance and the open-set performance essentially in a decoupling manner. As a comparison, only if $(h, r)$ makes correct predictions on both close-set samples and open-set samples, it will be free from the punishment of OpenAUC. In light of this, we have the following proposition, whose details will be presented in Appendix.C.4.

**Proposition 7.** *Given two close-set samples $(\boldsymbol{x}_1, y_1)$ and $(\boldsymbol{x}_2, y_2)$ and an open-set sample $(\boldsymbol{x}_3, C+1)$, if $(\tilde{h}, \tilde{r})$ makes the same predictions as $(h, r)$ expect that $h(\boldsymbol{x}_1) = \tilde{h}(\boldsymbol{x}_1) = y_1, h(\boldsymbol{x}_2), \tilde{h}(\boldsymbol{x}_2) \neq y_2, r(\boldsymbol{x}_2) > r(\boldsymbol{x}_3) > r(\boldsymbol{x}_1), r(\boldsymbol{x}_3) = \tilde{r}(\boldsymbol{x}_3)$ and $\tilde{r}(\boldsymbol{x}_2) = r(\boldsymbol{x}_1), \tilde{r}(\boldsymbol{x}_1) = r(\boldsymbol{x}_2)$, we have $\texttt{OpenAUC}(\tilde{h}, \tilde{r}) < \texttt{OpenAUC}(h, r)$.*

**Remark 8.** *If we construct $(\tilde{h}, \tilde{r})$ in the way that leads to the inconsistency property III, $(\tilde{h}, \tilde{r})$ will perform inferior to $(h, r)$ on OpenAUC, that is, OpenAUC is free from the inconsistency property III.*

To sum up, OpenAUC is free from the aforementioned inconsistency properties. In view of this, it has become an appealing problem to design learning methods that can optimize OpenAUC efficiently.

### 4.4 Learning Method for OpenAUC

Following the standard machine learning paradigm [27], we should first reformulate the metric optimization problem to a risk minimization problem. For traditional metrics such as the close-set accuracy, one can simply replace $\mathbf{1}[\mathcal{A}]$ with $\mathbf{1}[\neg\mathcal{A}]$. However, due to the product term, it is clear that

$$\mathcal{R}(f, r) := 1 - \texttt{OpenAUC} \neq \underset{\substack{\boldsymbol{z}_k \sim D_k \\ \boldsymbol{z}_u \sim D_u}}{\mathbb{E}} \left[ \mathbf{1}\left[y_k \neq h(\boldsymbol{x}_k)\right] \cdot \mathbf{1}\left[r(\boldsymbol{x}_u) \leq r(\boldsymbol{x}_k)\right] \right]. \tag{13}$$

To address this issue, we reformulate the product term to a simpler summation term, whose proof is presented in Appendix.C.5.

**Proposition 8.** *Optimizing OpenAUC is equivalent to the following risk minimization problem:*

$$\min_{f,r} \mathcal{R}(f, r) = \underset{\substack{\boldsymbol{z}_k \sim D_k \\ \boldsymbol{z}_u \sim D_u}}{\mathbb{E}} \left[ \underbrace{\mathbf{1}\left[y_k \neq h(\boldsymbol{x}_k)\right]}_{(a)} + \underbrace{\mathbf{1}\left[y_k = h(\boldsymbol{x}_k)\right]}_{(b)} \cdot \underbrace{\mathbf{1}\left[r(\boldsymbol{x}_u) \leq r(\boldsymbol{x}_k)\right]}_{(c)} \right] \tag{14}$$

**Remark 9.** *If we view term (b) as a switch of term (c), then $\mathcal{R}(f, r)$ essentially is a conditional combination of the close-set error and the AUC risk, that is, term (a) and term (c) respectively. In other words, we first minimize the close-set error, and only if $h$ makes a correct prediction, we will optimize the open-set AUC risk. This intuition is consistent with an OSR model's decision process described in Sec.2.*

On top of this observation, we have the following empirical minimization objective:

$$\hat{\mathcal{R}}_{L,\ell}(f, r) := \frac{1}{N_k} \sum_{i=1}^{N_k} L(h(\boldsymbol{x}_i), y_i) + \frac{1}{N_k N_u} \sum_{i=i}^{N_k} \sum_{j=1}^{N_u} \left[ \mathbf{1}\left[y_i = h(\boldsymbol{x}_i)\right] \cdot \ell(r(\boldsymbol{x}_j) - r(\boldsymbol{x}_i)) \right], \tag{15}$$

where $L$ is a common close-set classification loss function such as the cross-entropy loss; $\ell$ represents a continuous surrogate loss for AUC optimization such as the square loss $\ell_{sq}(t) = (1 - t)^2$, whose details can be found in the recent survey [21, 28, 29]; $\mathbf{1}[y_k = h(\boldsymbol{x}_k)]$ acts as the switch of $\ell$ and will not be updated in each epoch. Note that there exist no open-set samples in the training set. Inspired by [11], we adopt manifold mixup [30] to generate open-set samples. Specifically, the convex combinations of samples from different known classes are regarded as open-set samples:

$$\boldsymbol{x}_u = \lambda_\beta f_{pre}(\boldsymbol{x}_i) + (1 - \lambda_\beta)f_{pre}(\boldsymbol{x}_j), y_i \neq y_j, \tag{16}$$

Table 2: The OpenAUC results on six benchmark datasets for OSR, where C+10, C+50 and CE+ represent **CIFAR+10**, **CIFAR+50** and Cross-Entropy+, respectively. The best and the runner-up method on each dataset are marked with red and blue, respectively.

| Method | MNIST | SVHN | CIFAR10 | C+10 | C+50 | TinyImageNet |
|--------|-------|------|---------|------|------|--------------|
| Softmax | 99.2±0.1 | 92.8±0.4 | 83.8±1.5 | 90.9±1.3 | 88.5±0.7 | 60.8±5.1 |
| GCPL [31] | 99.1±0.2 | 93.4±0.6 | 84.3±1.7 | 91.0±1.7 | 88.3±1.1 | 59.3±5.3 |
| RPL [26] | 99.4±0.1 | 93.6±0.5 | 85.2±1.4 | 91.8±1.2 | 89.6±0.9 | 53.2±4.6 |
| ARPL [13] | 99.4±0.1 | 94.0±0.6 | 86.6±1.4 | 93.5±0.8 | 91.6±0.4 | 62.3±3.3 |
| ARPL+CS [13] | 99.5±0.1 | 94.3±0.3 | 87.9±1.5 | 94.7±0.7 | 92.9±0.3 | 65.9±3.8 |
| CE+ [14] | 99.1±0.2 | 93.9±0.4 | 88.1±1.7 | 93.2±0.6 | 90.2±0.4 | 74.3±3.9 |
| Acc+AUC | 99.3±0.2 | 94.0±0.9 | 87.6±1.9 | 93.6±1.0 | 92.0±0.5 | 74.0±4.0 |
| Ours | 99.4±0.1 | 95.0±0.4 | 89.2±1.9 | 95.2±0.7 | 93.6±0.3 | 75.9±4.1 |

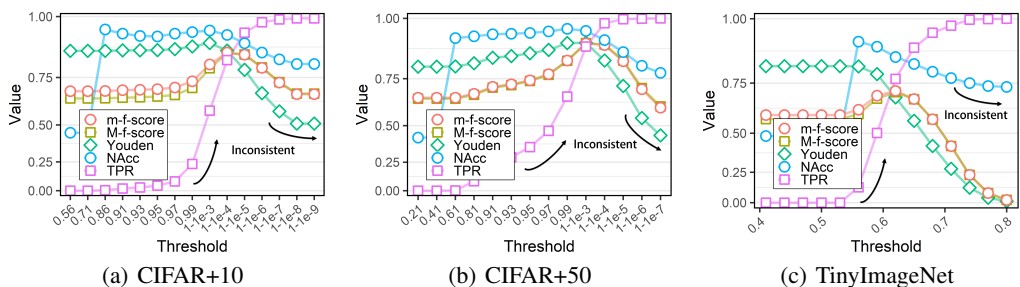

| (a) CIFAR+10 | (b) CIFAR+50 | (c) TinyImageNet |
|---|---|---|

Figure 1: The inconsistency property of classification-based metrics. We can find that all these metrics decrease rapidly as the TPR performance of unknown classes increases.

where $\boldsymbol{x}_i$ and $\boldsymbol{x}_j$ are samples from different known classes; $f$ is decomposed as $f_{post}(f_{pre}(\boldsymbol{x}))$; $\lambda_\beta \in (0,1)$ is sampled from a Beta distribution $B(\alpha,\alpha)$. Then, the score of $\boldsymbol{x}_u$ is obtained by $f_{post}(\boldsymbol{x}_u)$. Compared with other generative models [3, 12, 16], manifold mixup enjoys a significant efficiency advantage. While a potential problem is that $\boldsymbol{x}_u$ might locate close to the manifolds of other known classes, *i.e.*, $\lambda_\beta f_{pre}(\boldsymbol{x}_i) + (1-\lambda_\beta)f_{pre}(\boldsymbol{x}_j) \approx f_{pre}(\boldsymbol{x}_k), k \neq i, j$. To this end, we set an extra hyperparameter $\lambda$ for the AUC risk, and the final objective becomes:

$$\hat{\mathcal{R}}_{L,\ell,\lambda}(f,r) := \frac{1}{N_k}\sum_{i=1}^{N_k} L(h(\boldsymbol{x}_i), y_i) + \frac{\lambda}{N_k N_u}\sum_{i=i}^{N_k}\sum_{j=1}^{N_u}\left[\mathbf{1}\left[y_i = h(\boldsymbol{x}_i)\right] \cdot \ell(r(\boldsymbol{x}_j) - r(\boldsymbol{x}_i))\right]. \quad (17)$$

## 5  Experiments

In this section, extensive experiments are conducted to answer the following research questions: **(Q1)** Does the inconsistency property exist in empirical results? **(Q2)** Does optimize OpenAUC avoid the inconsistency properties and thus help the model outperform the state-of-the-art methods? **(Q3)** How does the proposed method perform under different hyperparameter settings?

### 5.1  Protocol

Following the protocol in [13] and [14], the experiments are conducted on the following datasets: (1) **MNIST**[1] [32], **SVHN**[2] [33] and **CIFAR10**[3] [34], where 4 classes are randomly sampled as the unknown classes; (2) **CIFAR+10** and **CIFAR+50**, where 4 classes are sampled from CIFAR10 as known classes, and $N$ non-overlapping classes sampled from the CIFAR100 dataset[3] [34] act as the unknown classes; (3) **TinyImageNet**[4] [35], where 20 and 180 classes are randomly sampled to evaluate the close-set and open-set performance; (4) **Fine-grained datasets** such as CUB[5] [36],

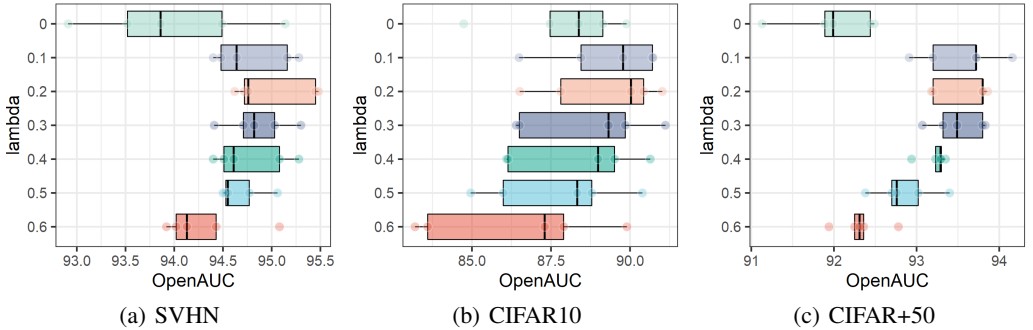

|  (a) SVHN | (b) CIFAR10 | (c) CIFAR+50 |

Figure 2: The sensitivity analysis of the proposed learning method for OpenAUC.

where known classes are subdivided into three disjoint sets {*Easy*, *Medium*, *Hard*} according to their semantic novelty.

To answer the question **(Q1)**, we record the model perform of Cross-Entropy+ [14] on open-set F-score, Youden' index, Normalized Accuracy, AUC, close-set accuracy and TPR of the open-set class. To answer the question **(Q2)**, we compare the proposed method with the state-of-the-art methods, including Softmax, GCPL [31], RPL [26], ARPL [13], ARPL+CS [13], Cross-Entropy+ [14]. To validate the inconsistency property III, we compare the objective that removes the term $\mathbf{1}\left[y_k = h(\boldsymbol{x}_k)\right]$ in Eq.(17), which is denoted by Acc+AUC. Besides, the implementation details are presented in Appendix.D

## 5.2 Results and Analysis

**Metric comparison (Q1).** We present the model performance on classification-based metrics under different thresholds in Fig.1. From these results, we can find that all the classification-based metrics decrease rapidly as the TPR performance of the unknown class increases. Moreover, on CIFAR+10 and TinyImageNet, the best NAcc performance corresponds to a very small TPR performance of the unknown class, which is consistent with our analysis in Prop.2. In other words, all these metrics encourage us selecting a threshold that misclassifies more the open-set samples. All these results show that the inconsistency properties do exist in practical scenarios.

**Performance Comparison (Q2).** We compare the proposed method with competitors in Tab.2 and have the following observations: (1) Our method outperforms the competitors consistently on all the datasets, which validate the effectiveness of the proposed learning method. (2) Acc+AUC performs inferior to the proposed method, which is consistent with the inconsistency property III. Besides, the results in fine-grained datasets are presented in Appendix.E.

**Sensitivity Analysis (Q3).** We present the sensitivity in terms of the hyperparameter $\lambda$ in Fig.2. From the results, we have the following observation: (1) Optimizing the OpenAUC risk consistently helps improve the model performance, which is consistent with our goal to design the OpenAUC objective. (2) When $\lambda$ equals 0.1 or 0.2, the proposed method achieves the best performance. (3) As $\lambda$ increases, there exists a decreasing trend in model performance, which might be induced by the noise in generated samples.

## 6 Broad Impact

This work provides a novel metric named OpenAUC for the OSR task, as well as its optimization method. We expect our research could promote the research of open-set learning, especially from the perspective of model evaluation. Moreover, no metric is perfect, and, of course, it is no exception

---

[1] http://yann.lecun.com/exdb/mnist/. Licensed GPL3.

[2] http://ufldl.stanford.edu/housenumbers/. Licensed GPL3.

[3] https://www.cs.toronto.edu/~kriz/cifar.html. Licensed MIT.

[4] http://cs231n.stanford.edu/tiny-imagenet-200.zip. Licensed MIT.

[5] https://www.vision.caltech.edu/datasets/cub_200_2011/. Licensed MIT.

for OpenAUC. To be specific, OpenAUC summarizes the COTPR performance *under all the OFPR performance*. However, some applications, such as self-driving, require a high recall of open-set. According to Prop.6, only the performance under low OFPR is of interest. In this case, OpenAUC might be biased due to considering irrelevant performances. This might cause potential negative impact concerning safety and security. To fix this issue, optimizing the partial OpenAUC, which summarizes the COTPR performance under some given OFPR performance, might be a better choice. Of course, there is no free lunch. Partial OpenAUC will be more difficult to optimize due to the selection operation. Besides, the generalization bound of open-set learning is still an opening problem, and we leave the corresponding analysis of OpenAUC optimization in future work.

## 7 Conclusion

This paper presents an extensive analysis of existing metrics for OSR. The theoretical results show that existing metrics are essentially inconsistent with the goal of OSR. To fix this issue, a novel metric named OpenAUC is proposed. Compared with existing metrics, OpenAUC enjoys a concise formulation that evaluates the close-set performance and open-set performance in a coupling manner. A series of propositions show that OpenAUC is free from the inconsistency properties of existing metrics. Finally, an end-to-end learning method is proposed for OpenAUC, and the experimental results validate the theoretical results and the effectiveness of the proposed method.

## Acknowledgments

This work was supported in part by the National Key R&D Program of China under Grant 2018AAA0102000, in part by National Natural Science Foundation of China: U21B2038, 61931008, U2001202, 61733007, 6212200758 and 61976202, in part by the Fundamental Research Funds for the Central Universities, in part by Youth Innovation Promotion Association CAS, in part by the Strategic Priority Research Program of Chinese Academy of Sciences, Grant No. XDB28000000, in part by the China National Postdoctoral Program for Innovative Talents under Grant BX2021298, and in part by China Postdoctoral Science Foundation under Grant 2022M713101.

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
