# Appendix

## A  Related Work of AUC Optimization

OpenAUC is naturally related to AUC [23] due to the pairwise formulation Eq.(12) and the surrogate loss used in Eq.(15). Specifically, for a binary classification problem, AUC, the Area Under the ROC Curve, measures the probability that the positive instances are ranked higher than the negative ones. Benefiting from this property, AUC is essentially insensitive to label distribution and thus has become a popular metric for imbalanced scenarios such as disease prediction [37] and novelty detection [18]. As pointed out in [23], optimizing the AUC performance cannot be realized by the traditional learning paradigm that minimizes the error rate. To this end, how to optimize the AUC performance has raised wide attention. In this direction, most early work focuses on the off-line setting [38, 39, 40]. And [28] provides a systematic analysis of the consistency property of common surrogate losses. Nowadays, more studies explore the online setting due to the rapid increase of the dataset scale [41, 42, 43, 44, 45], whose details can be found in the recent survey [21].

## B  Proof for the Inconsistency Property

### B.1  Proof for Proposition 1

**Proposition 1** (Inconsistency Property I). *Given a dataset $\mathcal{S}$ and a metric $\mathbb{M}$ that is invariant to $TP_{C+1}$, $FN_{C+1}$ and $FP_{C+1}$, then for any $(h, R)$ such that $\sum_{i=1}^{C} FP_i(h, R) \geq TP_{C+1}(h, R)$, there exists $(\tilde{h}, \tilde{R})$ such that $\mathbb{M}(\tilde{h}, \tilde{R}) = \mathbb{M}(h, R)$ but $TP_{C+1}(\tilde{h}, \tilde{R}) = 0$.*

*Proof.* We first consider a simpler case where only two predictions differ between $(h, R)$ and $(\tilde{h}, \tilde{R})$. As shown in Fig.3(a), given an open-set sample $(\boldsymbol{x}_1, C + 1)$ and a close-set sample $(\boldsymbol{x}_2, y_2)$, where $y_2 \neq C + 1$, if $(\tilde{h}, \tilde{R})$ makes the same predictions as $(h, R)$ expect that $R(\boldsymbol{x}_1) = 1, \tilde{R}(\boldsymbol{x}_1) = 0, \tilde{h}(\boldsymbol{x}_1) = h(\boldsymbol{x}_2)$ and $R(\boldsymbol{x}_2) = 0, h(\boldsymbol{x}_2) \neq y_2, \tilde{R}(\boldsymbol{x}_2) = 1$, it is not difficult to check that

- $\widetilde{FN}_{y_2} = FN_{y_2}$ since both $(h, R)$ and $(\tilde{h}, \tilde{R})$ fail to classify $\boldsymbol{x}_2$;

- $\widetilde{FP}_{h(\boldsymbol{x}_2)} = FP_{h(\boldsymbol{x}_2)}$ since $\boldsymbol{x}_1$ is misclassified as $h(\boldsymbol{x}_2)$, and $(\tilde{h}, \tilde{R})$ changes the prediction of $\boldsymbol{x}_2$ to $C + 1$ (increase and decrease $\widetilde{FP}_{h(\boldsymbol{x}_2)}$ by 1, respectively).

Under such a construction, we can find that $\mathbb{M}(\tilde{h}, \tilde{R}) = \mathbb{M}(h, R)$ but $\widetilde{TP}_{C+1} = TP_{C+1} - 1$. As long as

$$\sum_{i=1}^{C} \mathrm{FP}_i(h, R) \geq \mathrm{TP}_{C+1}(h, R),$$

we can utilize this construction iteratively until $\mathrm{TP}_{C+1}(\tilde{h}, \tilde{R}) = 0$, that is, $\tilde{R}(\boldsymbol{x}) \equiv 0$ for any $y = C + 1$.  $\square$

### B.2  Proof for Proposition 2

**Proposition 2** (Inconsistency Property II). *Given a dataset $\mathcal{S}$, for any classifier-rejector pair $(h, R)$ such that $\sum_{i=1}^{C} FN_i(h, R) \geq TP_{C+1}(h, R)$ and $TP_{C+1}(h, R) > FP_{C+1}(h, R)$, there exists $(\tilde{h}, \tilde{R})$ such that $\mathbb{N}Acc(\tilde{h}, \tilde{R}) > \mathbb{N}Acc(h, R)$ but $TP_{C+1}(\tilde{h}, \tilde{R}) = 0$.*

*Proof.* Similarly, we first consider a simpler case where only two predictions differ between $(h, R)$ and $(\tilde{h}, \tilde{R})$. As shown in Fig.3(b), given an open-set sample $(\boldsymbol{x}_1, C + 1)$ and a close-set sample $(\boldsymbol{x}_2, y_2)$, where $y_2 \neq C + 1$, if $(\tilde{h}, \tilde{R})$ makes the same predictions as $(h, R)$ expect that $R(\boldsymbol{x}_1) = 1, \tilde{R}(\boldsymbol{x}_1) = 0, \tilde{h}(\boldsymbol{x}_1) = y_2$ and $R(\boldsymbol{x}_2) = 1, \tilde{R}(\boldsymbol{x}_2) = 0, \tilde{h}(\boldsymbol{x}_2) = y_2$, we can find that

- $\widetilde{TN}_{y_2} = TN_{y_2} - 1, \widetilde{FP}_{y_2} = FP_{y_2} + 1$ and $\widetilde{TP}_{C+1} = TP_{C+1} - 1$ since $(\tilde{h}, \tilde{R})$ changes the prediction of $\boldsymbol{x}_1$ to $y_2$.

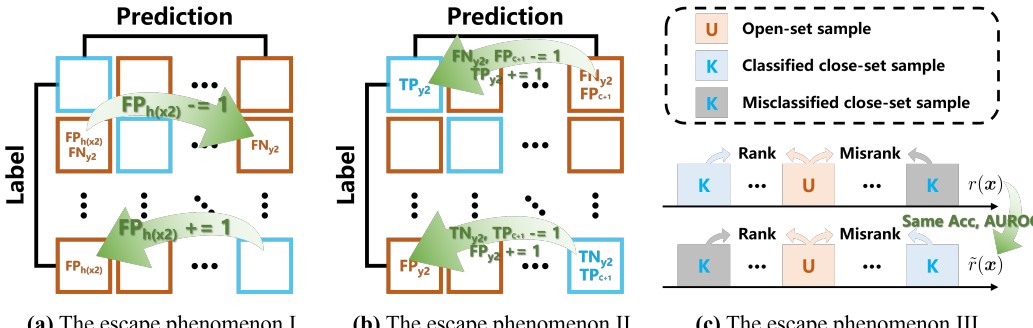

**(a)** The escape phenomenon I      **(b)** The escape phenomenon II      **(c)** The escape phenomenon III

Figure 3: The illustration of three inconsistency phenomena: **(a)** The metric invariant to $\text{TP}_{C+1}$, $\text{FN}_{C+1}$ and $\text{FP}_{C+1}$ suffers from the inconsistency property I, such as `F-score` and Youden's index; **(b)** `NA` suffers from the inconsistency property II; **(c)** the metric simply aggregating $\text{Acc}_k$ and `AUC` suffers from the inconsistency property III.

- $\widetilde{TP}_{y_2} = TP_{y_2} + 1$, $\widetilde{FN}_{y_2} = FN_{y_2} - 1$ and $\widetilde{FP}_{C+1} = FP_{C+1} - 1$ since $(\tilde{h}, \tilde{R})$ make a correct prediction on $\boldsymbol{x}_2$.

On one hand, we have $\text{AKS}(\tilde{h}, \tilde{R}) = \text{AKS}(h, R)$ since $\widetilde{TP}_{y_2} + \widetilde{TN}_{y_2} = TP_{y_2} + TN_{y_2}$ and $\widetilde{TP}_{y_2} + \widetilde{TN}_{y_2} + \widetilde{FP}_{y_2} + \widetilde{FN}_{y_2} = TP_{y_2} + TN_{y_2} + FP_{y_2} + FN_{y_2}$. On the other hand,

$$
\begin{aligned}
\text{AUS}(\tilde{h}, \tilde{R}) - \text{AUS}(h, R) &= \frac{\text{TP}_{C+1} - 1}{\text{TP}_{C+1} + \text{FP}_{C+1} - 2} - \frac{\text{TP}_{C+1}}{\text{TP}_{C+1} + \text{FP}_{C+1}} \\
&= \frac{\text{TP}_{C+1} - \text{FP}_{C+1}}{(\text{TP}_{C+1} + \text{FP}_{C+1} - 2)(\text{TP}_{C+1} + \text{FP}_{C+1})} \\
&> 0.
\end{aligned}
\tag{18}
$$

Consequently, we have $\text{NAcc}(\tilde{h}, \tilde{R}) > \text{NAcc}(h, R)$ but $\text{TP}_{C+1}(\tilde{h}, \tilde{R}) = \text{TP}_{C+1}(h, R) - 1$. As long as

$$
\sum_{i=1}^{C} \text{FN}_i(h, R) \geq \text{TP}_{C+1}(h, R),
$$

we can utilize this construction iteratively until $\text{TP}_{C+1}(\tilde{h}, \tilde{R}) = 0$, that is, $\tilde{R}(\boldsymbol{x}) \equiv 0$ for any $y = C + 1$. $\qquad\square$

### B.3 Proof for Proposition 3

**Proposition 3** (Inconsistency Property III). *Given a dataset $\mathcal{S}$, for any $(h, r)$ satisfying $\text{Acc}_k, \text{AUC} \neq 1$, there exists $(\tilde{h}, \tilde{r})$ that performs worse on the OSR task but satisfies:*

$$
agg(\text{Acc}_k(\tilde{h}), \text{AUC}(\tilde{r})) = agg(\text{Acc}_k(h), \text{AUC}(r)),
$$

*where $agg : \mathbb{R} \times \mathbb{R} \to \mathbb{R}$ is the aggregation function.*

*Proof.* Similarly, we first consider a simpler case where only two predictions differ between $(h, r)$ and $(\tilde{h}, \tilde{r})$. As shown in Fig.3(c), given two close-set samples $(\boldsymbol{x}_1, y_1), (\boldsymbol{x}_2, y_2)$ and an open-set sample $(\boldsymbol{x}_3, C + 1)$, if $(\tilde{h}, \tilde{r})$ makes the same predictions as $(h, r)$ expect that $h(\boldsymbol{x}_1) = \tilde{h}(\boldsymbol{x}_1) = y_1, h(\boldsymbol{x}_2), \tilde{h}(\boldsymbol{x}_2) \neq y_2, r(\boldsymbol{x}_2) > r(\boldsymbol{x}_3) > r(\boldsymbol{x}_1), r(\boldsymbol{x}_3) = \tilde{r}(\boldsymbol{x}_3)$ and $\tilde{r}(\boldsymbol{x}_2) = r(\boldsymbol{x}_1), \tilde{r}(\boldsymbol{x}_1) = r(\boldsymbol{x}_2)$, we can find that

- $\text{Acc}_k(\tilde{h}) = \text{Acc}_k(h)$ since $(\tilde{f}, \tilde{r})$ does not change the predictions on close-set;

- $\text{AUC}_k(\tilde{r}) = \text{AUC}_k(r)$ since the ordering between close-set samples and open-set samples is also not changed.

Consequently, we have $agg(\texttt{Acc}_k(\tilde{h}), \texttt{AUC}(\tilde{r})) = agg(\texttt{Acc}_k(h), \texttt{AUC}(r))$. However, $(\tilde{h}, \tilde{r})$ performs inferior to $(h, r)$ on the OSR task. To be specific, according to the prediction process described in Sec.2, if we select $t = r(\boldsymbol{x}_3)$, then $(h, r)$ will make correct predictions on $\boldsymbol{x}_1$ and $\boldsymbol{x}_3$. As a comparison, $(\tilde{h}, \tilde{r})$ only correctly classifies $\boldsymbol{x}_3$. In this case, the simple aggregation of $\texttt{Acc}_k$ and $\texttt{AUC}_k$ is clearly inconsistent with the model performance. $\square$

## C   Proof for OpenAUC

### C.1   Proof for Proposition 4

**Proposition 4.** *Given $(h, r)$ and a sample pair $(\boldsymbol{z}_k, \boldsymbol{z}_u), \boldsymbol{z}_k \in \mathcal{Z}_k, \boldsymbol{z}_u \in \mathcal{Z}_u$, OpenAUC equals the probability that $h$ makes correct prediction on $\boldsymbol{z}_k$ and $r$ ranks $\boldsymbol{z}_u$ higher than $\boldsymbol{z}_k$:*

$$\texttt{OpenAUC} = \underset{\substack{\boldsymbol{z}_k \sim D_k \\ \boldsymbol{z}_u \sim D_u}}{\mathbb{E}} \Big[ \underbrace{\mathbf{1}\left[y_k = h(\boldsymbol{x}_k)\right]}_{(a)} \cdot \underbrace{\mathbf{1}\left[r(\boldsymbol{x}_u) > r(\boldsymbol{x}_k)\right]}_{(b)} \Big]. \tag{12}$$

Let $X_1, X_0$ be continuous random variables for a close-set/open-set sample given by $r$. Let $Y \in \{0, 1\}$ be the random variable where $Y = 1$ means that the classifier $h$ makes a correct prediction for a close-set sample. Then, Let $f_1$ and $f_0$ be the density of $X_1|Y = 1$ and $X_2$ respectively and $F_1$ and $F_0$ be the cumulative distribution function of $X_1$ and $Y = 1, X_2$ respectively. Given a classifier $h$, an open-set score function $r$ and a threshold $t$, we have

$$\begin{aligned} \texttt{COTPR}(t) &= \mathbb{P}\left[X_1 \le t, Y = 1\right] = \mathbb{P}\left[X_1 \le t|Y = 1\right] \mathbb{P}\left[Y = 1\right] = F_1(t)\mathbb{P}\left[Y = 1\right], \\ \texttt{OFPR}(t) &= \mathbb{P}\left[X_0 \le t\right] = F_0(t). \end{aligned} \tag{19}$$

Then let $t = \texttt{OFPR}^{-1}(t)$, that is, $t = \texttt{OFPR}(t)$, we have

$$\begin{aligned} \texttt{OpenAUC} &= \int_{-\infty}^{+\infty} \texttt{COTPR}(t)\texttt{OFPR}'(t)\, dt \\ &= \int_{-\infty}^{+\infty} F_1(t)\mathbb{P}\left[Y = 1\right] f_0(t)\, dt \\ &= \mathbb{P}\left[Y = 1\right] \int_{-\infty}^{+\infty} F_1(t) f_0(t)\, dt \\ &= \mathbb{P}\left[Y = 1\right] \mathbb{P}\left[X_0 > X_1|Y = 1\right] \\ &= \mathbb{P}\left[X_0 > X_1, Y = 1\right]. \end{aligned} \tag{20}$$

Thus, we have

$$\texttt{OpenAUC}(f, r) = \underset{\substack{\boldsymbol{z}_k \sim D_k \\ \boldsymbol{z}_u \sim D_u}}{\mathbb{E}} \left[\mathbf{1}\left[y_k = h(\boldsymbol{x}_k)\right]\right] \cdot \mathbf{1}\left[r(\boldsymbol{x}_u) > r(\boldsymbol{x}_k)\right]. \tag{21}$$

### C.2   Proof for Proposition 5

**Proposition 5.** *Given a sample pair $((\boldsymbol{x}_1, C + 1), (\boldsymbol{x}_2, y_2))$, where $y_2 \ne C + 1$, for any $(h, r)$ such that $R(\boldsymbol{x}_1) = 1, R(\boldsymbol{x}_2) = 0, h(\boldsymbol{x}_2) \ne y_2$, if $(\tilde{h}, \tilde{r})$ makes the same predictions as $(h, r)$ expect that $\tilde{R}(\boldsymbol{x}_1) = 0, \tilde{h}(\boldsymbol{x}_1) = h(\boldsymbol{x}_2)$ and $\tilde{R}(\boldsymbol{x}_2) = 1$, we have $\texttt{OpenAUC}(\tilde{h}, \tilde{r}) < \texttt{OpenAUC}(h, r)$.*

*Proof.* Since $R(\boldsymbol{x}_1) = 1, \tilde{R}(\boldsymbol{x}_1) = 0$, we have $r(\boldsymbol{x}_1) > \tilde{r}(\boldsymbol{x}_1)$. According to the definition of OpenAUC,

$$\begin{aligned} &\texttt{OpenAUC}(h, r) - \texttt{OpenAUC}(\tilde{h}, \tilde{r}) \\ &= \frac{1}{N_k N_u} \sum_{i=1}^{N_k} \mathbf{1}\left[y_i = h(\boldsymbol{x}_i)\right] \left(\mathbf{1}\left[r(\boldsymbol{x}_1) > r(\boldsymbol{x}_i)\right] - \mathbf{1}\left[\tilde{r}(\boldsymbol{x}_1) > r(\boldsymbol{x}_i)\right]\right) \\ &= \frac{1}{N_k N_u} \sum_{i=1}^{N_k} \mathbf{1}\left[y_i = h(\boldsymbol{x}_i)\right] \cdot \mathbf{1}\left[r(\boldsymbol{x}_1) > r(\boldsymbol{x}_i) > \tilde{r}(\boldsymbol{x}_1)\right] \\ &\ge 0. \end{aligned} \tag{22}$$

Note that the equality holds only if there exists no correctly-classified close-set sample between $r(\boldsymbol{x}_1)$ and $\tilde{r}(\boldsymbol{x}_1)$. On one hand, this condition is not mild. On the other hand, in this case, $(\tilde{h}, \tilde{r})$ essentially shares the same OSR performance with $(h, r)$ since ranking open-set samples lower than misclassified closed-set samples does not affect model performance on the OSR task. In a word, we can conclude that $\texttt{OpenAUC}(\tilde{h}, \tilde{r}) < \texttt{OpenAUC}(h, r)$ as long as the inconsistency property II happens. $\qquad\square$

### C.3  Proof for Proposition 6

**Proposition 6.** *Given a dataset $\mathcal{S}$, for any $(f, r)$ such that $\texttt{OpenAUC} = k$ and any threshold $t_{C+1}$ such that $\texttt{FPR}_{C+1} = a \neq 0$, we have $\texttt{TPR}_{C+1} \geq 1 - (1 - k)/a$.*

*Proof.* Let $N_k$ and $N_u$ denote the number of close-set samples and open-set samples. According to the definition of OpenAUC, we have

$$1 - \texttt{OpenAUC} \geq 1 - \texttt{AUC} \geq \frac{\texttt{FP}_{C+1} \cdot \texttt{FN}_{C+1}}{N_u N_k}, \tag{23}$$

where the second inequlity holds the number of mis-ranked pair is greater than $\texttt{FP}_{C+1} \cdot \texttt{FN}_{C+1}$. Then, since

$$\texttt{FPR}_{C+1} = \frac{\texttt{FP}_{C+1}}{N_k}, \texttt{TPR}_{C+1} = \frac{\texttt{TP}_{C+1}}{N_u} = \frac{N_u - \texttt{FN}_{C+1}}{N_u}, \tag{24}$$

we have

$$1 - k \geq \texttt{FPR}_{C+1} \cdot (1 - \texttt{TPR}_{C+1}). \tag{25}$$

Finally,

$$\texttt{TPR}_{C+1} \geq 1 - (1 - k)/a. \tag{26}$$

$\qquad\square$

### C.4  Proof for Proposition 7

**Proposition 7.** *Given two close-set samples $(\boldsymbol{x}_1, y_1)$ and $(\boldsymbol{x}_2, y_2)$ and an open-set sample $(\boldsymbol{x}_3, C+1)$, if $(\tilde{h}, \tilde{r})$ makes the same predictions as $(h, r)$ expect that $h(\boldsymbol{x}_1) = \tilde{h}(\boldsymbol{x}_1) = y_1, h(\boldsymbol{x}_2), \tilde{h}(\boldsymbol{x}_2) \neq y_2, r(\boldsymbol{x}_2) > r(\boldsymbol{x}_3) > r(\boldsymbol{x}_1), r(\boldsymbol{x}_3) = \tilde{r}(\boldsymbol{x}_3)$ and $\tilde{r}(\boldsymbol{x}_2) = r(\boldsymbol{x}_1), \tilde{r}(\boldsymbol{x}_1) = r(\boldsymbol{x}_2)$, we have $\texttt{OpenAUC}(\tilde{h}, \tilde{r}) < \texttt{OpenAUC}(h, r)$.*

*Proof.* As shown in Fig.3(c), we have

$$
\begin{aligned}
\texttt{OpenAUC}&(h, r) - \texttt{OpenAUC}(\tilde{h}, \tilde{r}) \\
&= \frac{1}{N_k N_u} \sum_{j=1}^{N_u} \mathbf{1}\left[r(\boldsymbol{x}_j) > r(\boldsymbol{x}_1)\right] - \mathbf{1}\left[r(\boldsymbol{x}_j) > \tilde{r}(\boldsymbol{x}_1)\right] \\
&= \frac{1}{N_k N_u} \sum_{j=1}^{N_u} \mathbf{1}\left[r(\boldsymbol{x}_j) > r(\boldsymbol{x}_1)\right] - \mathbf{1}\left[r(\boldsymbol{x}_j) > r(\boldsymbol{x}_2)\right] \\
&= \frac{1}{N_k N_u} |\{(\boldsymbol{x}, C+1) : r(\boldsymbol{x}_2) > r(\boldsymbol{x}) > r(\boldsymbol{x}_1)\}| \\
&\geq 1.
\end{aligned} \tag{27}
$$

Then, we can conclude that $\texttt{OpenAUC}(\tilde{h}, \tilde{r}) < \texttt{OpenAUC}(h, r)$. $\qquad\square$

### C.5  Proof for Proposition 8

**Proposition 8.** *Optimizing OpenAUC is equivalent to the following risk minimization problem:*

$$\min_{f,r} \mathcal{R}(f, r) = \underset{\substack{\boldsymbol{z}_k \sim D_k \\ \boldsymbol{z}_u \sim D_u}}{\mathbb{E}} \left[ \underbrace{\mathbf{1}\left[y_k \neq h(\boldsymbol{x}_k)\right]}_{(a)} + \underbrace{\mathbf{1}\left[y_k = h(\boldsymbol{x}_k)\right]}_{(b)} \cdot \underbrace{\mathbf{1}\left[r(\boldsymbol{x}_u) \leq r(\boldsymbol{x}_k)\right]}_{(c)} \right] \tag{14}$$

Table 3: The truth table for the proof of Proposition 8.

| $I_k$ | $I_u$ | $1 - I_k \cdot I_u$ | $\neg I_k + I_k \cdot \neg I_u$ |
|-------|-------|---------------------|----------------------------------|
| 1 | 1 | 0 | 0 |
| 1 | 0 | 1 | 1 |
| 0 | 1 | 1 | 1 |
| 0 | 0 | 1 | 1 |

*Proof.* Given any pair of close-open sample pair $(\boldsymbol{x}_k, \boldsymbol{x}_u)$, let $I_k$ and $I_u$ indicate whether the events $y_k = h(\boldsymbol{x}_k)$ and $r(\boldsymbol{x}_u) > r(\boldsymbol{x}_k)$ happen, respectively. Then, according to the definition of the OpenAUC risk, we have

$$\mathcal{R}(f, r) = \mathop{\mathbb{E}}_{\substack{\boldsymbol{z}_k \sim D_k \\ \boldsymbol{z}_u \sim D_u}} \left[ 1 - I_k \cdot I_u \right]. \tag{28}$$

Meanwhile, the right-hand side of Eq.(14) can be denoted as

$$\mathop{\mathbb{E}}_{\substack{\boldsymbol{z}_k \sim D_k \\ \boldsymbol{z}_u \sim D_u}} \left[ \neg I_k + I_k \cdot \neg I_u \right]. \tag{29}$$

Then the proof completes by Tab.3. $\qquad\square$

# D   Implementation details

**Infrastructure.** All the experiments are carried out on an ubuntu server equipped with Intel(R) Xeon(R) Silver 4110 CPU and an Nvidia(R) TITAN RTX GPU. We implement the codes via `python` (v-3.8.11), and the main third-party packages include `pytorch` (v-1.9.0) [46], `numpy` (v-1.20.3) [47], `scikit-learn` (v-0.24.2) [48] and `torchvision` (v-0.10.1) [49].

**Backbone and Optimization Method.** We adopt the widely-used VGG32 model as the backbone [1, 2, 5, 13, 14], expect that ResNet50 [50] are utilized in CUB. Besides, the score function follows Def.1. According to the empirical results in [14], we train the model with a batch size of 128 for 600 epochs except TinyImageNet and CUB, for which we use 64 and 32, respectively. Meanwhile, we adopt an initial learning rate of 0.1 for all datasets except TinyImageNet and CUB, for which we use 0.01 and 0.001, respectively. We train with a cosine annealed learning rate, restarting the learning rate to the initial value at epochs 200 and 400. Besides, the RandAugment and label smoothing strategy provided by [14] is utilized for all experiments.

**Generation of Open-set Samples.** As elaborated in Sec.4.4, we utilize manifold mixup to generate open-set samples. Specifically, we first shuffle the received batch $B$, which produces a mini-batch $B'$. Then, mixup is conducted on the pairs in $B \times B'$, where $\times$ denotes *pointwise* product of two sets. Finally, the metric is calculated on the pairs in $B \times \tilde{B}$, where $\tilde{B}$ is the batch generated by the mixup operation. Note that we expect the instances in each pair from $B \times B'$ to have different class labels, so that the mixup examples (i.e., $\tilde{B}$) can be located somewhere outside the close-set domain. Hence, we eliminate the pairs from the same classes. Note that we only mixup the pairs at the same slot of $B$ and $B'$, and the metric is evaluated on the pairs at the same slot of $B$ and $\tilde{B}$. Hence, the time complexity is $O(|B|)$, rather than $O(|B|^2)$. According to the empirical results in [11], we set $\alpha = 2$ as the default value. Meanwhile, the hyperparameter $\lambda$ is searched in $\{0.1, 0.2, 0.3, 0.4, 0.5, 0.6\}$.

**Efficient Caculation of OpenAUC.** During the test phase, open-set samples are available. Benefiting from the pairwise formulation, we can calculate OpenAUC efficiently. Specifically, we first mask each close-set sample $\boldsymbol{x}_k$ that has been misclassified on the close-set. Specifically, we have

$$\tilde{r}(\boldsymbol{x}_k) \leftarrow \begin{cases} \epsilon + \max_{\boldsymbol{x}_u \in \mathcal{S}_u} r(\boldsymbol{x}_u), & h(\boldsymbol{x}_k) \neq y_k \\ r(\boldsymbol{x}_k), & \text{otherwise} \end{cases}$$

where $\mathcal{S}_u$ denotes the open-set, and $\epsilon > 0$ is a small constant. In this way, we have

$$\mathtt{OpenAUC}(f, r) = \frac{1}{N_k N_u} \sum_{i=1}^{N_k} \sum_{j=1}^{N_u} \mathbb{I}[y_i = h(\boldsymbol{x}_i)] \cdot \mathbb{I}[r(\boldsymbol{x}_j) > r(\boldsymbol{x}_i)]$$

$$= \frac{1}{N_k N_u} \sum_{i=1}^{N_k} \sum_{j=1}^{N_u} \mathbb{I}[\tilde{r}(\boldsymbol{x}_j) > \tilde{r}(\boldsymbol{x}_i)]$$

$$= \mathtt{AUC}(\tilde{r}).$$

In other words, OpenAUC degenerates to the traditional AUC, and common tools such as `scikit-learn` can boost the computation.

## E More empirical results

In this section, we present the empirical results on fine-grained datasets. For a comprehensive evaluation, we provide the model performances on multiple metrics such as Close-set Accuracy, AUC, OpenAUC, Error Rate@95%TPR, and Open-set F-score. All the results are recorded in Tab.4-5, where (E/M/H) corresponds to the results on the Easy/Medium/Hard split of open-set classes. Note that we report the open-set F-score under the optimal threshold. Besides, we did not analyze Error Rate@95%TPR in Sec.3 since it is a metric for novelty detection, and little OSR work adopted it as a metric. From the results, we have the following observations:

- The proposed method outperforms the competitors on novelty-detection metrics such as AUC and Error Rate@95%TPR, especially on the Medium and Hard splits. Moreover: (1) The improvement on AUC comes from the AUC-based term in the proposed objective, which is consistent with our theoretical expectation. (2) The result on Error Rate validates Prop.6 that optimizing Open-AUC reduces the upper bound of FPR. Recall that

$$ErrorRate \downarrow = 1 - Acc \uparrow = 1 - \frac{TP + TN}{TP + TN + FP + FN},$$

$$TPR = \frac{TP}{TP + FN}, TNR \uparrow = 1 - FPR \downarrow = \frac{TN}{TN + FP}.$$

- Our method achieves comparable performances on the close-set accuracy and Open-set F-score. This result is reasonable since compared with CE+, no more optimization is conducted on the close-set samples in our new objective function.

- Benefiting from the improvement on open-set samples and the comparable performance on close-set samples, the proposed method achieves the best performance on OpenAUC.

- Another observation is that the Open-set F-score shares similar values for all difficulty splits. Note that the only difference among these splits comes from their open-set data. This phenomenon shows that Open-set F-score cannot differentiate the performance on the open-set. This is inevitable since this metric evaluates the open-set performance only in an implicit manner. Hence, it again validates the necessity to adopt OpenAUC as the evaluation metric.

To sum up, the empirical results on fine-grained datasets again speak to the efficacy of OpenAUC and the proposed optimization method.

Table 4: Empirical results on CUB, where E/M/H corresponds to the results on the Easy/Medium/Hard split of open-set samples. The best and the runner-up method on each metric are marked with **red** and **blue**, respectively.

|  | Close-set Accuracy | AUC (E/M/H) | OpenAUC (E/M/H) |
|---|---|---|---|
| Softmax | 78.1 | 79.7 / 73.8 / 66.9 | 67.2 / 63.0 / 57.8 |
| GCPL [31] | 82.5 | 85.0 / 78.7 / 73.4 | 74.7 / 70.3 / 66.7 |
| RPL [26] | 82.6 | 85.5 / 78.1 / 69.6 | 74.5 / 69.0 / 62.4 |
| ARPL [13] | 82.1 | 85.4 / 78.0 / 70.0 | 74.4 / 68.9 / 62.7 |
| CE+ [14] | 86.2 | 88.3 / 82.3 / 76.3 | 79.8 / 75.4 / 70.8 |
| ARPL+ [14] | 85.9 | 83.5 / 78.9 / 72.1 | 76.0 / 72.4 / 66.8 |
| Ours | 86.2 | 88.8 / 83.2 / 78.1 | 80.2 / 76.1 / 72.5 |

Table 5: Empirical results on CUB, where E/M/H corresponds to the results on the Easy/Medium/Hard split of open-set samples. The best and the runner-up method on each metric are marked with **red** and **blue**, respectively.

|  | Error@95%TPR (E/M/H) | macro F-score (E/M/H) | micro F-score (E/M/H) |
|---|---|---|---|
| Softmax | 46.6 / 55.9 / 62.8 | 67.4 / 66.5 / 66.6 | 69.0 / 68.9 / 70.8 |
| GCPL [31] | 37.0 / 46.8 / 51.3 | 77.6 / 75.4 / 74.0 | 78.4 / 76.8 / 77.4 |
| RPL [26] | 39.5 / 53.5 / 64.0 | 75.4 / 73.3 / 72.4 | 76.7 / 75.2 / 76.6 |
| ARPL [13] | 37.6 / 49.9 / 62.7 | 75.3 / 73.1 / 72.2 | 76.6 / 75.0 / 76.5 |
| CE+ [14] | 28.4 / 42.1 / 52.3 | 82.6 / 80.3 / 78.3 | 83.3 / 81.6 / 81.4 |
| ARPL+ [14] | 48.7 / 60.6 / 67.8 | 80.8 / 79.0 / 77.3 | 81.7 / 80.4 / 80.4 |
| Ours | 28.1 / 39.7 / 47.6 | 82.2 / 79.7 / 78.1 | 83.0 / 81.2 / 81.1 |