# OpenReview forum: "OpenAUC: Towards AUC-Oriented Open-Set Recognition"
_NeurIPS.cc/2022/Conference — NeurIPS 2022 Accept_

### Official Review · Reviewer_tgk6 · 2022-07-05

**Rating:** 6
**Confidence:** 4
**Soundness:** 3 good
**Presentation:** 3 good
**Contribution:** 2 fair

**Summary:**

The paper introduces a new metric called OpenAUC as a summary number to jointly measure the closed-set classification accuracy and open-set detection accuracy. It is a threshold-free metric. The paper compares existing metrics used in the literature of open-set recognition and points out their limitations. Further, the paper develops a loss to train neural networks by directly optimizing the OpenAUC metric. Experiments on standard open-set recognition datasets validate the effectiveness of the loss.



**Questions:**

The paper is above average. As for questions, authors are encouraged to address the weaknesses listed above. Answers in the rebuttal can sway the rating.

**Limitations:**

The paper does not effectively discuss limitations and potential negative societal impacts. The design of the new metric is a summary number that is based on the heuristics (a model should be good at both close-set classification and open-set detection). There are indeed potential limitations and negative impacts. For example, such a summary number makes it non-trivial to select operating points in real-world systems and hide critical failures in the real world. Some applications require high recall of the open-set (e.g., autonomous vehicles) and some other favor high accuracy on the close-set (e.g., image tagging).

**Strengths And Weaknesses:**

Strengths:

- The motivation of the paper is good. It is desired to design a metric for open-set recognition.

- The paper nicely analyzes existing metrics and explain why they fail to jointly measure closed-set and open-set accuracies.

- The developed metric and the derived loss function make sense.


Weaknesses:

- One important weakness is that the paper does not justify why OpenAUC is better than OSCR. As pointed out in the paper (Line189), calculating the area under the OSCR curve can be a summary number (as done by [13,14]).

- Following the above, Line188 argues that "a numeric metric is generally necessary for model comparison." This argument conflicts that in [4] -- quote here: "The application of any algorithm to a real world problem involves the selection of an operating point, with the natural choices on a PR curve being either high precision (low number of false positives) or high recall (high number
of true positives)." That said, a curve offers richer information and allows comparing methods at different operating points. The paper should discuss this.

- While the derived loss helps train a model that directly optimizes OpenAUC, it should evaluate the trained model using other metrics as well, including OSCR, F-measure, OSCR, closed-set accuracy, and AUROC. Admittedly, these metrics (except OSCR) have issues, given the standard datasets, using all these metrics give a balanced understanding -- it is important to justify the proposed loss function helps, or doesn't decrease much, the accuracies of closed-set classification and open-set detection.

- It is not clear what the final open-set function r is in experiments.

- Line 90 and 91: Is it a typo "close-set score function r"? Isn't r an open-set score function as stated in Line86?

---

> ### Author Response · Authors · 2022-08-01
> **The response to reviewer tgk6 (Part 3/3)**
>
> > Comment (5): Line 90 and 91: Is it a typo "close-set score function r"? Isn't r an open-set score function as stated in Line86?
>
> **Ans**: Thanks for your careful reading, and the answer is positive. We are sorry for these typos and will correct them in the future version.
>
> > Comment (6): Limitations and potential negative societal impacts of OpenAUC.
>
> **Ans**: Thanks very much for these constructive concerns! OpenAUC summarizes the OTPR performance under *all* the OFPR performance. However, as described in the comment, some applications require a high recall of open-set (i.e., $TPR_{C+1}$). According to Prop.6, only the performance under low OFPR is of interest. In this case, OpenAUC might be biased due to considering irrelevant performances. This might cause potential negative societal impacts. To fix this issue, optimizing the partial OpenAUC, which summarizes the OTPR performance under some given OFPR performance, might be a better choice. Of course, there is no free lunch. Partial OpenAUC will be more difficult to optimize due to the selection operation. Meanwhile, the comment points out that some other applications might favor close-set performance. We argue that OpenAUC is free from this concern since the product formulation of OpenAUC requires that the close-set samples are correctly classified. In other words, a low close-set accuracy will inevitably induce a low OpenAUC. We will update these discussions in the future version.

---

> > ### Comment · Reviewer_tgk6 · 2022-08-08
> > **further comments**
> >
> > Thanks for the rebuttal.
> >
> > Because of the typo of r, I didn't follow the open-set function r. Now I think I understand. In plain language, I think the open-set function r is exactly the same as "1 - max-of-softmax-score", right? If so, it is natural to ask whether other methods (other than max-of-softmax) such as max-of-logit [14] benefits from the OpenAUC optimized network. I encourage the authors to run this simple experiment.
> >
> > Moreover, I encourage the authors to have a well-thought-out discussion in the next version about (1) OpenAUC vs. OSCR w.r.t curves and summary number, (2) operating point for real-world applications and limitations of openAUC.
> >
> > A small complaint -- NeurIPS allows authors to update manuscript during rebuttal, and doing so helps build trust between authors and reviewers. I suggest authors do so next time. I encourage authors to release code as well.
> >
> > I maintain my rating as weak accept.

---

> > > ### Author Response · Authors · 2022-08-08
> > > **Response to the further comments**
> > >
> > > Thank you very much for your nice comments! we would like to make the following responses.
> > >
> > > > Comment (a): The open-set function r is the exactly same as "1 - max-of-softmax-score", right? If so, it is natural to ask whether other methods such as max-of-logit [14] benefit from the OpenAUC optimized network.
> > >
> > > **Ans**: Thank you very much for this constructive comment! Both answers are positive. We conduct this experiment on the CUB dataset. As shown in the following table, CE+, the max-of-logit method, also benefits from the proposed optimization objective.
> > >
> > > | CUB     | Close Acc | AUC(E/M/H)*           | OpenAUC(E/M/H)        | TNR@95(E/M/H)         | macro-F(E/M/H) | micro-F(E/M/H) |
> > > |:------- |:---------:|:---------------------:|:---------------------:|:---------------------:|:-------:|:-------:|
> > > | CE+           | 86.2     | 88.3 / 82.3 / 76.3 | 79.8 / 75.4 / 70.8 | 28.4 / 42.1 / 52.3 | 82.6 / 80.3 / 78.3  | 83.3 / 81.6 / 81.4 |
> > > | CE+OpenAUC    | 86.1     | 88.7 / 82.9 / 77.6 | 80.1 / 75.8 / 72.0 | 27.8 / 39.5 / 46.7 | 81.9 / 79.6 / 77.7  | 82.5 / 81.1 / 80.8 |
> > >
> > > > Comment (b): Moreover, I encourage the authors to have a well-thought-out discussion in the next version about (1) OpenAUC vs. OSCR w.r.t curves and summary number, (2) operating point for real-world applications and limitations of openAUC.
> > >
> > > **Ans**: Thank you very much for this suggestion! We have updated the responses to the latest version. For the sake of your convenience, we attach the corresponding parts right here.
> > >
> > > For (a), i.e., Comment (1) and (2):
> > >
> > > **Sec.4.1, Line 188-194.** Compared with the aforementioned metrics, the OSCR curve contains richer information and allows comparing model performances at different operating points. While our goal is to optimize the overall performance of the curve. Hence, it is necessary to find a numeric metric that aggregates the information of the entire curve, such that the models can be trained by optimizing the loss of the metric. To this end, [13] and [14] estimate the area under the OSCR curve by directly calculating the numeric integral with histograms. However, this number is hard to optimize due to multiple non-differential operators such as ranking and counting.
> > >
> > > **Sec. 4.2, Line 220-222.** Moreover, Compared with the OSCR curve, OpenAUC enjoys a concise formulation, based on which we can design a differentiable objective function for Empirical Risk Minimization (ERM). We will present the details in Sec.4.4.
> > >
> > > For (b), i.e., Comment (6):
> > >
> > > **Sec.6 Broad Impact.** This work provides a novel metric named OpenAUC for the OSR task, as well as its optimization method. We expect our research could promote the research of open-set learning, especially from the respective of model evaluation. Moreover, no metric is perfect, and, of course, it is no exception for OpenAUC. To be specific, OpenAUC summarizes the OTPR performance *under all the OFPR performance*. However, some applications, such as self-driving, require a high recall of open-set. According to Prop.6, only the performance under low OFPR is of interest. In this case, OpenAUC might be biased due to considering irrelevant performances. This might cause potential negative impact concerning safety and security. To fix this issue, optimizing the partial OpenAUC, which summarizes the OTPR performance under some given OFPR performance, might be a better choice. Of course, there is no free lunch. Partial OpenAUC will be more difficult to optimize due to the selection operation. Besides, the generalization bound of open-set learning is still an opening problem, and we leave the corresponding analysis of OpenAUC optimization in future work.
> > >
> > > Besides, (1) New empirical results are attached in Appendix.D, as well as our observations. (2) The typos in Line 90 and 91 have been revised. (3) The final open-set function is highlighted in Appendix.C.
> > >
> > >
> > > > Comment (c): Update the manuscript.
> > >
> > > **Ans**: Thanks for this constructive suggestion! We have updated the manuscript according to the comments, and all the revisions are marked with blue.

---

> ### Author Response · Authors · 2022-08-01
> **The response to reviewer tgk6 (Part 2/3)**
>
> > Comment (3): It is important to justify the proposed loss function helps, or doesn't decrease much, the accuracies of closed-set classification and open-set detection.
>
> **Ans**: Thanks very much for your constructive suggestion! We conduct an additional experiment on a more challenging dataset, i.e., CUB [14]. According to your suggestion, we present the model performances on multiple metrics such as Close-set Accuracy, AUC, OpenAUC, Error Rate@95%TPR, and Open-set F-score. Note that we report the open-set F-score under the optimal threshold. Besides, we did not analyze Error Rate@95%TPR in Sec.3 since it is a metric for novelty detection, and little OSR work adopted it as a metric. For your convenience, the new results are attached as follows, where (E/M/H) corresponds to the results on the Easy/Medium/Hard split of open-set classes.
>
> | CUB     | Close Acc | AUC(E/M/H)*           | OpenAUC(E/M/H)        | Error@95(E/M/H)         | macro-F(E/M/H) | micro-F(E/M/H) |
> |:------- |:---------:|:---------------------:|:---------------------:|:---------------------:|:-------:|:-------:|
> | Softmax | 78.1     | 79.7 / 73.8 / 66.9 | 67.2 / 63.0 / 57.8 | 46.6 / 55.9 / 62.8 | 67.4 / 66.5 / 66.6  | 69.0 / 68.9 / 70.8 |
> | GCPL    | 82.5     | 85.0 / 78.7 / 73.4 | 74.7 / 70.3 / 66.7 | 37.0 / 46.8 / 51.3 | 77.6 / 75.4 / 74.0  | 78.4 / 76.8 / 77.4 |
> | RPL     | 82.6     | 85.5 / 78.1 / 69.6 | 74.5 / 69.0 / 62.4 | 39.5 / 53.5 / 64.0 | 75.4 / 73.3 / 72.4  | 76.7 / 75.2 / 76.6 |
> | ARPL    | 82.1     | 85.4 / 78.0 / 70.0 | 74.4 / 68.9 / 62.7 | 37.6 / 49.9 / 62.7 | 75.3 / 73.1 / 72.2  | 76.6 / 75.0 / 76.5 |
> | CE+     | 86.2     | 88.3 / 82.3 / 76.3 | 79.8 / 75.4 / 70.8 | 28.4 / 42.1 / 52.3 | 82.6 / 80.3 / 78.3  | 83.3 / 81.6 / 81.4 |
> | ARPL+   | 85.9     | 83.5 / 78.9 / 72.1 | 76.0 / 72.4 / 66.8 | 48.7 / 60.6 / 67.8 | 80.8 / 79.0 / 77.3  | 81.7 / 80.4 / 80.4 |
> | Ours    | 86.2     | 88.8 / 83.2 / 78.1 | 80.2 / 76.1 / 72.5 | 28.1 / 39.7 / 47.6 | 82.2 / 79.7 / 78.1  | 83.0 / 81.2 / 81.1 |
>
> From the results, we have the following observations:
>
> - The proposed method outperforms the competitors on novelty-detection metrics such as AUC and Error Rate@95%TPR, especially on the Medium and Hard splits. Moreover: (1) The improvement on AUC comes from the AUC-based term in the proposed objective, which is consistent with our theoretical expectation. (2) The result on Error Rate validates Prop.6 that optimizing Open-AUC reduces the upper bound of FPR (Recall that $Error Rate \downarrow = 1 - Acc \uparrow = 1 - \frac{TP + TN}{TP + TN + FP + FN}, TPR = \frac{TP}{TP + FN}, TNR \uparrow =  1 - FPR \downarrow = \frac{TN}{TN + FP}$).
>
> - Our method achieves comparable performances on the close-set accuracy and Open-set F-score. This result is reasonable since compared with CE+, no more optimization is conducted on the close-set samples in our new objective function.
>
> - Benefiting from the improvement on open-set samples and the comparable performance on close-set samples, the proposed method achieves the best performance on OpenAUC.
>
> - Another observation is that the Open-set F-score shares similar values for all difficulty splits. Note that the only difference among these splits comes from their open-set data. This phenomenon shows that Open-set F-score cannot differentiate the performance on the open-set. This is inevitable since this metric evaluates the open-set performance only in an implicit manner. Hence, it again validates the necessity to adopt OpenAUC as the evaluation metric.
>
> To sum up, the empirical results on CUB again speak to the efficacy of OpenAUC and the proposed optimization method. We will update these results in the next version.
>
> > Comment (4): It is not clear what the final open-set function r is in experiments.
>
> **Ans**: Perhaps due to the way of our writing, it is a pity to leave the impression that the open-set function is not well formulated. As described in line 195-197, the final open-set function is defined as $\min_{k \in \mathcal{Y}_k} f(\boldsymbol{x}_k)$, where $\forall c \in \mathcal{Y}_k, f(\boldsymbol{x}_c) \propto \mathbb{P}[y \neq c | \boldsymbol{x}]$. We will highlight this fact in the future version.

---

> ### Author Response · Authors · 2022-08-01
> **The response to reviewer tgk6 (Part 1/3)**
>
> Thank you for your comments! We would like to make the following response:
>
> > Comment (1): Why OpenAUC is better than calculating the area under the OSCR?
>
> **Ans**: Thank you very much for this constructive comment! We agree with the reviewer that [13,14] also get a numerical metric by calculating the area under the OSCR curve. While the point here is *the way to finish the calculation*. Our final goal is to find a reasonable OSR objective function to optimize directly. To this end, we need to get a simplified version of the metric, so that we can design a differentiable objective function for ERM (Empirical Risk Minimization). The existing studies [13,14] estimate the area under the OSCR curve by directly calculating the numerical integral with histograms, which involves multiple non-differential operators such as ranking and counting. By contrast, OpenAUC can be expressed as the sum of pair-wise loss terms, which enjoys a similar form to AUC. Inspired by the ERM framework of AUC, we can easily construct a differentiable objective function to optimize OpenAUC. For completeness, we will update these discussions in the future version.
>
> > Comment (2): The argument in Line188 conflicts with that in [4].
>
> **Ans**: Thanks for your nice question! We've realized that our expression might have induced some misunderstanding. In fact, our argument does not conflict with that in [4]. We also agree that the operation curve contains richer information than a single metric. However, again, our goal is to optimize the overall performance of the curve. Hence, we have to find a numerical metric that aggregates the information of the entire curve, so that the models can be trained by optimizing the loss of the metric. In this sense, our metric is necessary since compatible with its corresponding OFPR-OTPR curve. We will clarify this issue in the future version.

---

### Official Review · Reviewer_kn16 · 2022-07-10

**Rating:** 9
**Confidence:** 5
**Soundness:** 4 excellent
**Presentation:** 3 good
**Contribution:** 4 excellent

**Summary:**

This paper focuses on the evaluation issue for the Open-Set Recognition (OSR) problem. Specifically, the authors point out that existing metrics for OSR are inconsistent with the goal of OSR: some poor open-set predictions can escape from the punishment of classification-based metrics, while novelty detection AUC ignores the close-set performance. In view of this, a novel metric, named OpenAUC, is proposed. Theoretical analysis reveals that OpenAUC overcomes the limitations of the existing metrics. Moreover, an end-to-end learning algorithm is proposed to optimize OpenAUC. Finally, empirical results on six benchmark datasets validate the effectiveness of the proposed method.

**Questions:**

In Appendix, Figure.3 specifies the indices of classes, that is, “FP1”, “FP2”, “FN1” and so on. How do these classes correspond to the notations in the proof such as $y_1$ and $y_2$? It might be a bit confusing, and more clarification is necessary.

**Limitations:**

Yes

**Strengths And Weaknesses:**

Strengths:
 The authors make a systematic analysis of existing metrics for Open-set Recognition. Concretely, existing metrics are grouped into classification-based ones and novelty-detection-based ones. And their limitations are collectively categorized into three types of inconsistency properties.
 A novel metric named OpenAUC is proposed for OSR. Compared with existing metrics, OpenAUC (1) evaluates the performance on close-set and open-set in a unified manner; (2) aggregates the performance under different thresholds; (3) enjoys a concise formulation and thus is easier to optimize.
 The empirical results are convincing. On one hand, it validates the inconsistency properties of existing metrics. On the other hand, the effectiveness of the proposed learning method is validated.

Weakness:
It is recommended that more related work can be provided, such as the details of the competitors and AUC optimization.
Besides, there exist some minor issues: It is recommended to use “Eq.(1)” instead of “Equation (1)”. Some punctuations are missing such as the full stops of Eq.(9) and Eq.(12).

---

> ### Author Response · Authors · 2022-08-01
> **The response to reviewer kn16**
>
> Thank you for your comments! We would like to make the following response:
>
> > Comment (1): More related work of the competitors and AUC optimization.
>
> **Ans**: Thank you for this nice suggestion! For detailed information, [a,b,c] might be good references. We will provide a brief review and add some latest literature in the future version.
>
> [a] Chuanxing Geng, Sheng-Jun Huang, Songcan Chen: Recent Advances in Open Set Recognition: A Survey. IEEE Trans. Pattern Anal. Mach. Intell. 43(10): 3614-3631 (2021)
>
> [b] Mohammadreza Salehi, Hossein Mirzaei, Dan Hendrycks, Yixuan Li, Mohammad Hossein Rohban, Mohammad Sabokrou: A Unified Survey on Anomaly, Novelty, Open-Set, and Out-of-Distribution Detection: Solutions and Future Challenges. CoRR abs/2110.14051 (2021)
>
> [c] Tianbao Yang, Yiming Ying: AUC Maximization in the Era of Big Data and AI: A Survey. CoRR abs/2203.15046 (2022)
>
> > Comment (2): Some minor typos.
>
> **Ans**: Thanks for your careful reading! We will correct these typos in the future version.
>
> > Comment (3): In Appendix, Figure.3 specifies the indices of classes, that is, “FP1”, “FP2”, “FN1” and so on. How do these classes correspond to the notations in the proof?
>
> **Ans**: We are sorry for this confusing presentation. For Fig.3(a), the indices "1" and "2" corresponds to $h(\boldsymbol{x}_2)$ and $y_2$, respectively. For Fig.3(b), the index "1" corresponds to $y_2$. We will improve the figure in the future version.

---

### Official Review · Reviewer_DygR · 2022-07-11

**Rating:** 6
**Confidence:** 5
**Soundness:** 4 excellent
**Presentation:** 4 excellent
**Contribution:** 3 good

**Summary:**

This paper introduced a new evaluation metric for OSR, OpenAUC, which jointly measures the binary open-set performance and multi-class closed-set performance, and also introduced a simple OSR method by minimizing the OpenAUC risk with synthetic open-set samples by mixing up features of closed-set samples. Thorough theoretical analysis is presented and promising performance on public datasets is shown.

**Questions:**

Please refer to the weaknesses above. In addition,
- The OpenAUC requires measuring pairs of closed and open set samples, which is different from existing methods that only measure individual data points.  How the pairs are generated during the evaluation? Are the same pairs used for all methods? The possible pairs will be way more than the individual data points? This will significantly increase the evaluation cost from O(n) to O(n^2), thus reducing efficiency.

**Limitations:**

Briefly mentioned the future work on generalization bound which is not considered at the moment for OpenAUC. No societal impact is discussed, and I didn't see any major concerns here.

**Strengths And Weaknesses:**

Strengths:
+ The joint evaluation of open-set and closed-set performance is an important problem for the OSR problem.
+ Thorough theoretical analysis is given for existing methods and the proposed method; the proposed method is simple and effective.
+ Good results are obtained on public datasets.

Weaknesses:
- The experimental datasets seem already quite saturated, except TinyImageNet. The more challenging SSB datasets, introduced in [14] with a particular focus on better evaluating OSR, should be evaluated, to strengthen the paper.
- Only the VGG32 backbone is used. It would be good to see how the proposed method works on other backbones as well, such as ResNet, ViT.
- Figure 1 is partly unclear to me. What is the difference between m-f-score and M-f-score? Why OSCR curves are not presented? OSCR is the most relevant metric, and the OSCR curves should be presented.

One suggestion (not weakness): when describing the weaknesses in the main text on other methods, it would be better to add some demonstrative figures to show some examples and how the proposed method can handle them.

Overall, I hold a positive view on the proposed method and think it can be helpful for future OSR research, especially evaluation, but the experiment parts need improvement, especially on the more challenging datasets.

---

> ### Author Response · Authors · 2022-08-01
> **The response to reviewer DygR (Part 3/3)**
>
> > Comment (6): Briefly mention the future work on generalization bound which is not considered at the moment for OpenAUC.
>
> **Ans**: Generalization analysis for OSR is an appealing but rather challenging direction. To be concrete, existing techniques for generalization analysis are mostly based on the assumption that the training set and the test set are sampled from the same distribution, while it becomes invalid in OSR. Similar challenges appear in the open-set domain adaptation (OSDA) and novelty detection, but all related results require the test samples to be available in the training phase [a,b]. To fix this issue, [c] makes a strong assumption on the open-set distribution and leverages an off-the-shelf result of the density ratio estimation method. But how to do it for other methods, in general, remains an opening question. Moreover, our main focus in this paper is to find a proper metric for OSR to guide optimization and training. The generalization analysis is out of our scope.
>
> [a] Si Liu, Risheek Garrepalli, Thomas G. Dietterich, Alan Fern, Dan Hendrycks: Open Category Detection with PAC Guarantees. ICML 2018: 3175-3184
>
> [b] Zhen Fang, Jie Lu, Feng Liu, Junyu Xuan, Guangquan Zhang: Open Set Domain Adaptation: Theoretical Bound and Algorithm. IEEE Trans. Neural Networks Learn. Syst. 32(10): 4309-4322 (2021)
>
> [c] Zhen Fang, Jie Lu, Anjin Liu, Feng Liu, Guangquan Zhang: Learning Bounds for Open-Set Learning. ICML 2021: 3122-3132
>
> > Comment (7): No societal impact is discussed.
>
> **Ans**: Thanks very much for this constructive concern! Indeed, no metric is perfect, and, of course, it is no exception for OpenAUC. Specifically, OpenAUC summarizes the OTPR performance under *all* the OFPR performance. However, in many practical scenarios such as self-driving, only the OTPR performance under *low* OFPR performance is of interest. There might be some negative societal impact concerning safety and security. In this case, OpenAUC might be biased due to considering irrelevant performances. To fix this issue, partial OpenAUC, which summarizes the OTPR performance under some given OFPR performance, might be a better choice. Of course, there is no free lunch. Partial OpenAUC will be more difficult to optimize due to the selection operation. So, we leave it as future work and will include this discussion in the next version.

---

> > ### Comment · Reviewer_DygR · 2022-08-08
> > **Thanks for the response and further comments**
> >
> > I appreciate the responses and my main concerns have been addressed properly. Hence, I have raised my rating. The complete evaluation on the SSB datasets is suggested to be included in the final version, which will better validate the performance and benefit future works down the line.

---

> > > ### Author Response · Authors · 2022-08-08
> > > **Thanks for your nice comment and updates in the revision**
> > >
> > > Thank you very much for your nice comment! We have updated the responses to the latest version. To be specific, (1) new empirical results are attached in Appendix.D, as well as our observations. (2) More implementation details, including the new backbone, the generation of open-set pairs, and the efficiency issue, can be found in Appendix.C. (3) Fig.1 has been revised to eliminate the confusing notation. (4) A new section, i.e., Sec.6 is attached to discuss the potential societal impact of OpenAUC. Moreover, the empirical results on all the SBB datasets will be updated in the final version.

---

> ### Author Response · Authors · 2022-08-01
> **The response to reviewer DygR (Part 2/3)**
>
> > Comment (3): Figure 1 is partly unclear to me. (a) What is the difference between m-f-score and M-f-score? (b) Why OSCR curves are not presented?
>
> **Ans**: Perhaps due to the way of our writing, it is a pity to leave Fig.1 partly confusing.
>
> For (a), as presented in Eq.(2) and Eq.(3), M-f-score and m-f-score represent Open-set F-score that aggregates Precision and Recall in a *macro* and *micro* manner, respectively. For the sake of your convenience, we attach the equations right here.
> $$ \texttt{F-score} := 2 \times \frac{\texttt{P}\_k \times \texttt{TPR}\_k}{\texttt{P}\_k + \texttt{TPR}\_k},$$
> where
> $$\texttt{P}\_k := \frac{1}{C} \sum\_{i=1}^{C} \frac{\texttt{TP}\_{i}}{\texttt{TP}\_{i}+\texttt{FP}\_{i}}, \texttt{TPR}\_k := \frac{1}{C} \sum\_{i=1}^{C} \frac{\texttt{TP}\_{i}}{\texttt{TP}\_{i}+\texttt{FN}\_{i}}$$
> if we aggregate model performances in a *macro* manner, and
> $$\texttt{P}\_k := \frac{\sum\_{i=1}^{C} \texttt{TP}\_{i}}{\sum\_{i=1}^{C}\left(\texttt{TP}\_{i}+\texttt{FP}\_{i}\right)}, \texttt{TPR}\_k := \frac{\sum\_{i=1}^{C} \texttt{TP}\_{i}}{\sum\_{i=1}^{C}\left(\texttt{TP}\_{i}+\texttt{FN}\_{i}\right)}$$
> if the performances are summarized in a *micro* manner. To eliminate confusion, these two metrics will be denoted as macro-F-score and micro-F-score in the next version, respectively.
>
> For (b), the reason is two-fold: On one hand, Fig.1 aims to illustrate the inconsistency property of F-score, Youden's index, and normalized accuracy, while the OSCR curve does not suffer from an inconsistency property. Moreover, in Fig.1, we plot metric values against thresholds. However, to plot the OSCR curve, we need to plot CCR against FPR. Hence, we cannot plot them in the same figure.
>
> > Comment (4): Suggestion (not weakness): show some demonstrative figures when describing the weaknesses of other methods.
>
> **Ans**: Thanks for this constructive suggestion! We have demonstrated some examples in Appendix (Fig.3). To make the propositions easier to understand, we will add more figures and attach them to the main text in the future version.
>
> > Comment (5): How the open-set pairs are generated during the evaluation? Are the same pairs used for all methods? The efficiency issue.
>
> **Ans**: Our response consists of the evaluation on the training set and the test set, respectively.
>
> - When training, open-set samples are unavailable, and thus we adopt the mixup strategy on each batch $B$ to generate open-set samples. Specifically, we shuffle the received batch, which produces a mini-batch $B'$, and then conduct mixup on the pairs in $B \times B'$, where $\times$ denotes *pointwise* product of two sets. Finally, the metric is calculated on the pairs in $B \times \tilde{B}$, where $\tilde{B}$ is the batch generated by the mixup operation. Note that we expect the instances in each pair from $B \times B'$ to have different class labels, so that the mixup examples (i.e., $\tilde{B}$) can be located somewhere outside the close-set domain. Hence, we eliminate the pairs from the same classes. Note that we only mixup the pairs at the same slot of $B$ and $B'$, and the metric is evaluated on the pairs at the same slot of $B$ and $\tilde{B}$. Hence, the time complexity is still $O(|B|)$. Empirically, the training time for 600 epochs increases from 16h23min to 16h25min, which is quite efficient. Besides, we fix the random seed to guarantee the same pairs are generated for all methods.
>
> - During the test phase, open-set samples are available. Benefiting from the pairwise formulation, we can calculate OpenAUC efficiently. Specifically, we first mask each close-set sample $\boldsymbol{x}\_k$ that has been misclassified on the close-set. Specifically, we have
> $$\tilde{r}(\boldsymbol{x}\_k) \gets \begin{cases}
> \epsilon + \max\_{\boldsymbol{x}\_u \in \mathcal{S}\_u} r(\boldsymbol{x}\_u),&  h(\boldsymbol{x}\_k) \neq y_k\\\\
> r(\boldsymbol{x}\_k),& \text{otherwise}
> \end{cases}$$
> where $\mathcal{S}\_u$ denotes the open-set, and $\epsilon > 0$ is a small constant. In this way, we have
> $$\begin{aligned}
>     \texttt{OpenAUC}(f, r) & = \frac{1}{N\_k N\_u} \sum\_{i=1}^{N\_k}\sum\_{j=1}^{N\_u}\mathbb{I}[y\_i = h(\boldsymbol{x}\_i)] \cdot \mathbb{I}[r(\boldsymbol{x}\_j) > r(\boldsymbol{x}\_i)] \\\\
>     & = \frac{1}{N\_k N\_u} \sum\_{i=1}^{N\_k}\sum\_{j=1}^{N\_u} \mathbb{I}[\tilde{r}(\boldsymbol{x}\_j) > \tilde{r}(\boldsymbol{x}\_i)] \\\\
>     & = \texttt{AUC}(\tilde{r}).
> \end{aligned}$$
> In other words, OpenAUC degenerates to the traditional AUC, and common tools such as Scikit-learn can boost the computation. Besides, the pairs are naturally the same for all methods.

---

> ### Author Response · Authors · 2022-08-01
> **The response to reviewer DygR (Part 1/3)**
>
> Thanks for your comments! We would like to make the following responses. For the sake of your convenience, new empirical results are first attached:
>
> | CUB     | Close Acc | AUC(E/M/H)*           | OpenAUC(E/M/H)        | Error@95(E/M/H)         | macro-F(E/M/H) | micro-F(E/M/H) |
> |:------- |:---------:|:---------------------:|:---------------------:|:---------------------:|:-------:|:-------:|
> | Softmax | 78.1     | 79.7 / 73.8 / 66.9 | 67.2 / 63.0 / 57.8 | 46.6 / 55.9 / 62.8 | 67.4 / 66.5 / 66.6  | 69.0 / 68.9 / 70.8 |
> | GCPL    | 82.5     | 85.0 / 78.7 / 73.4 | 74.7 / 70.3 / 66.7 | 37.0 / 46.8 / 51.3 | 77.6 / 75.4 / 74.0  | 78.4 / 76.8 / 77.4 |
> | RPL     | 82.6     | 85.5 / 78.1 / 69.6 | 74.5 / 69.0 / 62.4 | 39.5 / 53.5 / 64.0 | 75.4 / 73.3 / 72.4  | 76.7 / 75.2 / 76.6 |
> | ARPL    | 82.1     | 85.4 / 78.0 / 70.0 | 74.4 / 68.9 / 62.7 | 37.6 / 49.9 / 62.7 | 75.3 / 73.1 / 72.2  | 76.6 / 75.0 / 76.5 |
> | CE+     | 86.2     | 88.3 / 82.3 / 76.3 | 79.8 / 75.4 / 70.8 | 28.4 / 42.1 / 52.3 | 82.6 / 80.3 / 78.3  | 83.3 / 81.6 / 81.4 |
> | ARPL+   | 85.9     | 83.5 / 78.9 / 72.1 | 76.0 / 72.4 / 66.8 | 48.7 / 60.6 / 67.8 | 80.8 / 79.0 / 77.3  | 81.7 / 80.4 / 80.4 |
> | Ours    | 86.2     | 88.8 / 83.2 / 78.1 | 80.2 / 76.1 / 72.5 | 28.1 / 39.7 / 47.6 | 82.2 / 79.7 / 78.1  | 83.0 / 81.2 / 81.1 |
>
> \*  E/M/H corresponds to the results on the Easy/Medium/Hard split of open-set classes.
>
> > Comment (1): The experimental datasets seem already quite saturated.
>
> **Ans**: Thank you very much for this constructive comment! We conduct an additional experiment on an SSB dataset, i.e., CUB. The above table presents the model performances on multiple metrics such as Close-set Accuracy, AUC, OpenAUC, Error Rate@95%TPR, and Open-set F-score, where (E/M/H) corresponds to the results on the Easy/Medium/Hard split of open-set classes. Note that we report the open-set F-score under the optimal threshold. From the results, we have the following observations:
>
> - The proposed method outperforms the competitors on novelty-detection metrics such as AUC and Error Rate@95%TPR, especially on the Medium and Hard splits. Moreover: (1) The improvement on AUC comes from the AUC-based term in the proposed objective, which is consistent with our theoretical expectation. (2) The result on Error Rate validates Prop.6 that optimizing Open-AUC reduces the upper bound of FPR (Recall that $Error Rate \downarrow = 1 - Acc \uparrow = 1 - \frac{TP + TN}{TP + TN + FP + FN}, TPR = \frac{TP}{TP + FN}, TNR \uparrow =  1 - FPR \downarrow = \frac{TN}{TN + FP}$).
>
> - Our method achieves comparable performances on the close-set accuracy and Open-set F-score. This result is reasonable since compared with CE+, no more optimization is conducted on the close-set samples in our new objective function.
>
> - Benefiting from the improvement on open-set samples and the comparable performance on close-set samples, the proposed method achieves the best performance on OpenAUC.
>
> - Another observation is that the Open-set F-score shares similar values for all difficulty splits. Note that the only difference among these splits comes from their open-set data. This phenomenon shows that Open-set F-score cannot differentiate the performance on the open-set. This is inevitable since this metric evaluates the open-set performance only in an implicit manner. Hence, it again validates the necessity to adopt OpenAUC as the evaluation metric.
>
> To sum up, the empirical results on CUB again speak to the efficacy of OpenAUC and the proposed optimization method. We will update these results in the next version.
>
> > Comment (2): It would be good to see how the proposed method works on other backbones.
>
> **Ans**: Thanks for this nice suggestion! The experiment above adopts ResNet-50 as the backbone. Due to the time limit, we can only finish the experiments on the CUB dataset. We will add the experiments for all the other datasets in the next version. We hope these results can make the proposed method more convincing.

---

### Meta-Review · Area_Chair_UdWF · 2022-08-21

**Recommendation:** Accept
**Confidence:** Certain

**Metareview:**

The paper proposes OpenAUC, which is a novel metric designed specifically for evaluating Open-Set Recognition (OSR) performance. OpenAUC is motivated by a formal analysis on existing OSR evaluation metrics, which suffer from three types of inconsistency properties. Theoretical results show that OpenAUC is consistent with the goal of OSR while free of all identified inconsistency properties. An empirical loss function is developed accordingly that enables model training to optimize the proposed OpenAUC.

Overall, the paper is well-written. The proposed OpenAUC metric can potentially benefit future research in OSR as recognized by the reviewers. Authors and reviewers engaged in a productive discussion, which helped to further improve the quality of the paper. The authors are encouraged to address some remaining suggestions from the reviewers, including adding results on other backbones in the experiments and extending the related work by discussing more recent literature in OSR and AUC optimization.


**Award:**

No

---

### Decision · Program_Chairs · 2022-09-14

Accept